# CALIBRATED INFORMATION BOTTLENECK
# FOR TRUSTED MULTI-MODAL CLUSTERING

**Shizhe Hu[1,2], Zhangwen Gou[1], Shuaiju Li[1], Jin Qin[1], Xiaoheng Jiang[1,2],**
**Pei Lv[1,2] & Mingliang Xu[1,2,3,4]** *

[1]School of Computer Science and Artificial Intelligence, Zhengzhou University
[2]Artificial Intelligence and Robotics Institute, Zhengzhou University
[3]Shandong Bosuan Zhixin Information Technology Co., Ltd
[4]State Key Laboratory of Metabolic Dysregulation & Prevention and Treatment
of Esophageal Cancer
ieshizhehu@gmail.com

## ABSTRACT

Information Bottleneck (IB) Theory is renowned for its ability to learn simple, compact, and effective data representations. In multi-modal clustering, IB theory effectively eliminates interfering redundancy and noise from multi-modal data, while maximally preserving the discriminative information. Existing IB-based multi-modal clustering methods suffer from low-quality pseudo-labels and over-reliance on accurate Mutual Information (MI) estimation, which is known to be challenging. Moreover, unreliable or noisy pseudo-labels may lead to an over-confident clustering outcome. To address these challenges, this paper proposes a novel CaLibrated Information Bottleneck (CLIB) framework designed to learn a clustering that is both accurate and trustworthy. We build a parallel multi-head network architecture—incorporating one primary cluster head and several modality-specific calibration heads—which achieves three key goals: namely, calibrating for the distortions introduced by biased MI estimation thus improving the stability of IB, constructing reliable target variables for IB from multiple modalities and producing a trustworthy clustering result. Notably, we design a dynamic pseudo-label selection strategy based on information redundancy theory to extract high-quality pseudo-labels, thereby enhancing training stability. Experimental results demonstrate that our model not only achieves competitive clustering accuracy on multiple benchmark datasets but also exhibits excellent performance on the expected calibration error metric. Code is available at https://shizhehu.github.io/.

## 1 INTRODUCTION

**Information Bottleneck Theory** (Tishby et al., 2000; Hu et al., 2024) provides an elegant theoretical framework for learning concise and effective data representations, centered on finding an optimal balance between data compression and information preservation. This is achieved by creating a compressed representation $T$ for an input $X$ that is optimized to retain features relevant to a target $Y$. The optimization is governed by the objective function:

$$\mathcal{L}_{min} = I(X;T) - \beta I(T;Y) \tag{1}$$

Where $I(\cdot;\cdot)$ denotes the mutual information. The parameter $\beta$ balances the trade-off between compression and relevance. By minimizing $I(X;T)$ while simultaneously maximizing $I(T;Y)$, IB is able to learn a compact yet informative representation of the input data. In the context of deep learning, the advent of methods such as the Deep Variational Information Bottleneck (Alemi et al., 2017) has enabled the integration of the IB principle with neural networks.

Since IB is capable of extracting pure features, it has been utilized in multi-modal clustering methods. However, applying the IB principle directly to multi-modal clustering faces a fundamental

---

*Corresponding author.

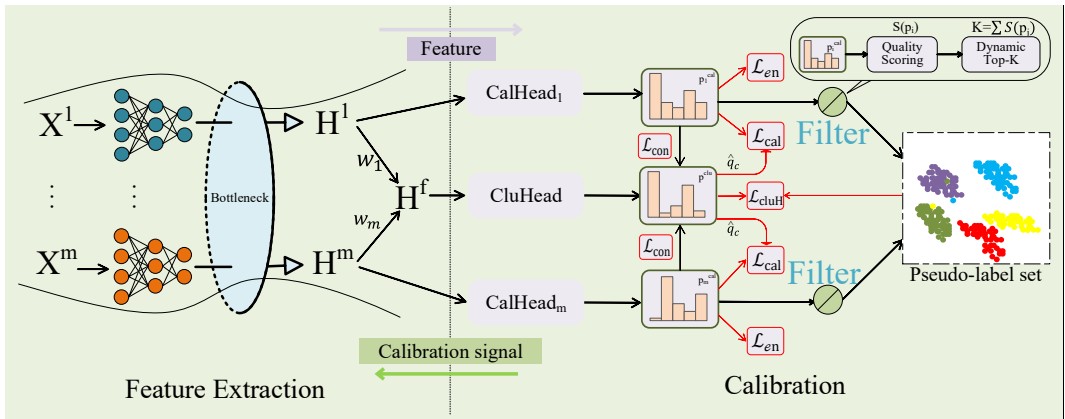

Figure 1: Illustration of the proposed CLIB framework. First, compact features $H^i$ are extracted from various modalities using IB. These multi-modal features are then integrated into a fused feature representation $H^f$. While the features from each modality serve as inputs to multiple calibration heads, the fused representation is processed by the cluster head. A screening mechanism is employed to select high-quality pseudo-labels from the output of calibration heads, which in turn supervise the training of the cluster head. This not only generates high-quality clustering results but also backpropagates a calibration signal to IB via the gradients from the cluster head, thereby guiding the learning of IB.

challenge: the efficacy of IB is highly dependent on a reliable target variable. In supervised learning, this target is typically the ground-truth label; in self-supervised learning, this role is filled by pseudo-labels, which act as the model's internal proxy for correctness. In current multi-modal clustering approaches utilizing IB (Hu et al., 2025; Yan et al., 2024), model-generated pseudo-labels are commonly used for guidance. A critical issue arises because these pseudo-labels are often noisy, particularly during the initial training stages. This creates a vicious cycle: low-quality pseudo-labels guide the learning of poor representations, which in turn produce flawed pseudo-labels for the next iteration. This feedback loop can amplify initial errors, causing the model to become progressively overconfident in its incorrect predictions. Therefore, a calibration mechanism is necessary to break this degenerative cycle and mitigate model overconfidence.

**The Computation of Mutual Information**, which measures the degree of association between two random variables, is critical to the performance of IB. However, calculating MI for high-dimensional, complex data (*e.g.*, image or text features) using its classical mathematical definition, which involves integration or summation over the true probability distributions, is extremely difficult, if not practically infeasible. Czyz et al. (2023) points out that existing estimators (Belghazi et al., 2018; Song & Ermon, 2020) are often evaluated only on simple data distributions (*e.g.*, multivariate normal distributions), which fails to reflect their performance on complex, real-world data, further underscoring the **difficulty of accurate MI estimation**. This difficulty poses a significant barrier to the effective application of the IB principle in complex, real-world scenarios.

To address these challenges, we propose a novel CLIB framework, which leverages the IB principle to learn compact representations from each modality. These representations are then integrated via an adaptive weighted mean fusion strategy to establish a unified cross-modal semantic space. Architecturally, as shown in Figure 1, CLIB consists of multiple modality-specific calibration heads and a single cluster head that operates on the fused representation. These heads engage in a reciprocal learning process, providing mutual supervision while strategically decoupling the calibration and final clustering objectives.

Notably, we introduce a pseudo-label selection mechanism based on information redundancy, quantified by entropy, to perceive the clustering quality of individual modalities and subsequently filter for high-confidence pseudo-labels to supervise the model's training. Furthermore, we adopt a two-stage training methodology: IB is first pre-trained to learn robust features before the calibration module is introduced, which significantly reduces the noise propagated into the calibration stage. This creates a symbiotic relationship: IB provides compact features for the downstream cluster-

ing task, while the calibration module offers corrective feedback through backpropagated gradients, thereby refining the feature extraction process itself. Crucially, this mechanism ensures that the feature learning is not misled by noisy pseudo-labels. To our knowledge, this is the first work addressing the trusted multi-modal clustering problem with calibrated IB framework.

**Our contributions** are threefold:

- We pioneer a method applying IB with calibrated, high-quality, and low-noise target variables which enhances the robustness of IB to extract compact and discriminative features from multi-modal data.

- Unlike existing IB methods, CLIB mitigates the over-reliance on precise MI estimation. Even if the MI estimated for a particular modality is biased, the calibration mechanism allows for its correction by leveraging information from other modalities, thereby enhancing the overall clustering performance of this framework. To our knowledge, this is the first work to introduce calibration for the performance issues in IB that arise from the inaccurate estimation of MI.

- Benefiting from the calibration mechanism, our proposed method not only achieves superior accuracy but also demonstrates a significantly lower Expected Calibration Error (ECE) compared to state-of-the-art multi-modal clustering methods. This enhances the trustworthiness of the IB framework and effectively alleviates the issue of model overconfidence.

## 2 RELATED WORK

### 2.1 MUTUAL INFORMATION ESTIMATION

To overcome the challenge of calculating MI for high-dimensional data, previous studies have predominantly adopted several strategies for approximation: Variational Inference-Based Approximation method (Barber & Agakov, 2004; Alemi et al., 2017) transforms the complex calculation of MI into an optimizable lower-bound estimation problem by introducing a parameterized variational distribution to approximate the true posterior distribution. The performance of this method is highly dependent on the expressive power of the chosen variational distribution, as an overly simplistic distribution family can result in an inaccurate estimation of MI. Neural Network-Based Estimation (Belghazi et al., 2018; Song & Ermon, 2020) utilizes the strong fitting capabilities of neural networks to estimate MI by learning its divergence form directly from samples via gradient descent, thus circumventing the need for explicit probability density estimation. In a mini-batch stochastic gradient descent training environment, the method's gradient estimates can be biased, requiring the introduction of bias-correction techniques to improve its stability and performance. Contrastive Learning-Based Estimation(van den Oord et al., 2018; Qiu et al., 2021) transforms the problem of MI estimation into a contrastive learning task, learning an effective lower bound by maximizing the similarity between "positive sample pairs" and minimizing it for "negative sample pairs". The performance of this approach is sensitive to the quantity and quality of negative samples, as an insufficient number of negative samples can impair the learning of discriminative features and thus reduce the accuracy of the MI estimation.

The aforementioned analysis reveals that all estimation methods for MI have certain drawbacks and inherent inaccuracies. In the context of IB-based multi-modal clustering, it is crucial to mitigate the effects of these potential inaccuracies to obtain more precise clustering results.

### 2.2 MULTI-MODAL CLUSTERING

Multi-modal data is characterized by complementary modalities, which provide distinct facets of information about the same object. In recent years, a series of multi-modal clustering methods has been proposed, including approaches that utilize Attention Mechanisms (Akbari et al., 2021; Huang et al., 2023), Adversarial Learning (Ganin et al., 2016). A class of approaches have focused on leveraging pseudo-labels in self-supervised frameworks. Pseudo-labels generated using fixed thresholds (Sohn et al., 2020; Gansbeke et al., 2020) often suffer from low quality and are riddled with noise. Consequently, they can misdirect the model's learning trajectory. This problem is exacerbated over iterative training, as the model becomes increasingly confident in its incorrect predictions—a process of error accumulation that results in a highly overconfident model.

Since IB is able to extract compact and discriminative features beneficial for clustering tasks, many IB-based clustering methods have been proposed. PTIB (Lou et al., 2025a) proposes a "peer-review" mechanism to learn modality weights in a parameter-free manner through mutual evaluation and trustworthiness scoring between modalities. SWIB (Lou et al., 2024) introduces self-supervised information from pseudo-labels into the weighted IB, learning weights by combining view quality with clustering consistency. MSDIB (Hu et al., 2025) proposes a "multi-aspect self-guided" strategy that comprehensively utilizes the private information, shared information, and pseudo-labels of modalities to learn cluster-friendly representations through the information bottleneck. Overall, due to its powerful representation learning capabilities, IB has been applied to various clustering algorithms in recent years. However, applying IB in an unsupervised setting requires the construction of high-quality, reliable target variables, which necessitates a mechanism for filtering pseudo-labels.

## 2.3 TRUSTWORTHY CLUSTERING

While deep clustering models have achieved significant performance, they often suffer from "over-confidence," where the model's predicted confidence far exceeds its actual accuracy(Jia et al., 2025). Accurate confidence estimation is crucial for building trustworthy decision-making systems, especially in safety-critical domains such as medical diagnosis (Mimori et al., 2021) and autonomous driving (Feng et al., 2019). PTIB (Lou et al., 2025a) achieves trustworthy clustering by emulating a "peer-review" mechanism, wherein different data modalities mutually evaluate one another to learn reliable fusion weights. AGARL (Yang et al., 2024) employs an "alternating generative adversarial" strategy to enforce the alignment of feature representations from different views, thereby learning a consistent and robust shared representation for clustering. TMC (Han et al., 2021) quantifies the uncertainty of each view by treating network outputs as "evidence" and subsequently fuses multi-view information based on evidence theory to obtain trustworthy results. Most existing confidence calibration techniques are not applicable to unsupervised clustering. For instance, post-processing methods like Temperature Scaling (Guo et al., 2017) require a labeled validation set, which is unavailable in clustering. Meanwhile, regularization techniques like Label Smoothing (Müller et al., 2019) excessively penalize high-confidence, reliable samples, preventing the model from effectively distinguishing between reliable and unreliable predictions. This gap highlights the need for calibration methods specifically designed for the unsupervised clustering paradigm.

## 3 PROPOSED METHOD

**Problem Definition and Overall Framework.** In the multi-modal clustering task, we are given a dataset $X = \{x_1, x_2, \ldots, x_N\}$ containing $N$ samples. Each sample $x_i$ is composed of data from $M$ different modalities, *i.e.*, $x_i = \{x_i^1, x_i^2, \ldots, x_i^M\}$, where $x_i^m$ represents the $m$-th modality for sample $i$. Our objective is to partition these $N$ samples into $C$ predefined clusters. In this section, we will provide a detailed introduction to our proposed CLIB framework. First, we describe how the IB principle is applied to learn both effective modality-specific and fused features. Then we will elaborate on the core of the framework: the calibration and clustering strategy, which consists of calibration heads and a cluster head, as well as a learning-aware strategy based on information redundancy for filtering high-quality pseudo-labels. The overall loss function for the framework is given below, where $\alpha$ is a parameter that balances the contributions of the feature extraction and the calibration strength.

$$\mathcal{L}_{total} = \mathcal{L}_{IB} + \alpha \mathcal{L}_{Cal} \qquad (2)$$

Notably, regarding pseudo-label screening strategies, Jia et al. (2025) and Li et al. (2022) utilize the maximum probability value of the distribution to dynamically select pseudo-labels, whereas this paper employs a pseudo-label screening method based on information redundancy theory.

## 3.1 REPRESENTATION LEARNING BASED ON INFORMATION BOTTLENECK

**Feature Extraction with Information Compression.** To extract compact and discriminative features from each modality, we have designed the loss function:

$$\mathcal{L}_{\text{C}} = \sum_{i=1}^{M} I\left(X^i, H^i\right) - \sum_{i=1}^{M} \sum_{j=i+1}^{M} I\left(H^i, H^j\right) \qquad (3)$$

Here, $X^i$ is the input for modality $i$, and $H^i$ is its learned representation. The first term, $I(X^i, H^i)$, serves as a compression objective. By minimizing it, we encourage the network to discard irrelevant information from the input. The second term, $-I(H^i, H^j)$, aims to maximize MI between representations of different modalities. This objective forces the unimodal encoders to learn a shared semantic space, a process often referred to as feature alignment. The alignment term ensures that in the process of compression, the model preserves semantic information that is common across modalities, which is assumed to be discriminative. Minimizing Equation 3 thus produces compact and aligned unimodal features.

**Information Preservation in Fusion.** To effectively integrate the features learned from different modalities, we employ an adaptive weighted-average fusion mechanism. This mechanism assigns a learnable weight $w_m$ to the representation of each modality $H^m$, and the final fused representation $H^f = w_1 H^1 + w_2 H^2 \cdots + w_M H^M$ with $\sum_{m=1}^{M} w_m = 1$. This adaptive mechanism allows the model to dynamically adjust the importance of different modalities based on their contribution to the final clustering task. The Information Preservation loss function is defined as:

$$\mathcal{L}_P = \sum_{m=1}^{M} I(H^m; H^f) \tag{4}$$

To ensure that the fused global representation retains the specific information from each modality, we maximize MI between each modal representation $H^m$ and the fused representation $H^f$. Combining the two parts above, the total information bottleneck loss function is defined as:

$$\mathcal{L}_{IB} = \mathcal{L}_C - \beta \mathcal{L}_P \tag{5}$$

Here, a larger value for $\beta$ places greater emphasis on the feature fusion process to yield a more comprehensive fused representation, albeit at the expense of unimodal feature purity. In contrast, a smaller $\beta$ is geared towards information compression to obtain more compact features, potentially at the cost of modality-specific fidelity in fused feature. Through the optimization of Equation 5, compact modality-specific features are obtained, along with fused features that sufficiently extract global common semantics.

Notably, because the features extracted in the early stages of training are still noisy and not fully compressed, prematurely introducing the calibration mechanism would allow a large amount of noise into the calibration module and propagate erroneous calibration signals, preventing the model from learning correctly. To address this, we have designed a two-stage training method that warms up IB, introducing the calibration module only after the learned features have become relatively stable.

## 3.2 CALIBRATION MECHANISM OF THE PARALLEL MULTI-HEAD ARCHITECTURE

After the warm-up stage of IB, the model has learned compact and effective features. The goal of the calibration mechanism is to use multi-head collaboration to resolve potential issues that could affect the final clustering performance, such as MI estimation bias and low-quality data in a specific modality, while also addressing the problem of model overconfidence. The overall loss of the calibration module could be:

$$\mathcal{L}_{Cal} = \mathcal{L}_{caliH} + \mathcal{L}_{en} + \mathcal{L}_{cluH} + \mathcal{L}_{con} \tag{6}$$

Where $\mathcal{L}_{caliH}$ and $\mathcal{L}_{cluH}$ optimizes the calibration heads and cluster head respectively, $\mathcal{L}_{re}$ serves as an entropy regularization term to prevent trivial solutions, and $\mathcal{L}_{con}$ is the consistency loss that forces the model to output a flat distribution when opinions conflict.

### 3.2.1 CALIBRATION HEADS

The primary task of the calibration heads is to learn a probability distribution for each single modality and to perceive the learning status of the current modality. A modality with a good learning status should theoretically produce a more discriminative clustering probability vector, which allows us to assess whether the modality has been sufficiently learned.

First, we run the K-Means algorithm on the fused feature space to partition the samples into $C$ pseudo-clusters, denoted as $Q_c$. Then, for each pseudo-cluster $Q_c$, we calculate the mean

of the output probabilities $p^{clu}$ from the cluster head for all samples within that cluster: $\hat{q}_c = \frac{1}{|Q_c|} \sum_{x_i \in Q_c} p_i^{clu}$. Here, $|Q_c|$ is the number of samples in pseudo-cluster $Q_c$. This cluster-averaging strategy leverages the neighborhood structure of samples in the feature space, effectively smoothing out the noise from individual sample predictions and thus providing a more stable and reliable supervisory signal for the training of the calibration heads. Subsequently, we optimize each calibration head by minimizing

$$\mathcal{L}_{caliH} = -\frac{1}{B} \sum_C \sum_{x_i \in Q_c} \sum_{m=1}^{M} \hat{q}_c \log(p_{i,m}^{cal}) \tag{7}$$

Where $B$ denotes the batch size, and $p_{i,m}^{cal}$ is the output probability from the calibration head for the $m$-th modality of sample $x_i$.

Furthermore, to prevent the calibration heads from producing trivial solutions during optimization an entropy regularization loss $\mathcal{L}_{re}$ is introduced:

$$\mathcal{L}_{re} = \frac{1}{M} \sum_{m=1}^{M} \bar{p}_m^{cal} \log(\bar{p}_m^{cal}) \tag{8}$$

The $\bar{p}_m^{cal}$ denotes the average predicted probability distribution over the entire batch for the $m$-th modality's calibration head. Equation 8 aims to encourage prediction diversity and avoid model collapse.

**Note.** As each calibration head acts as a modality-specific "expert" that only observes single-modality features, which may contain noise or conflicting information between modalities. For this reason, we stop the gradient backpropagation from the calibration heads to the bottleneck. Not doing so would require the bottleneck to satisfy multiple, potentially contradictory objectives simultaneously, which could lead to training instability or a confused direction in feature learning. This is not merely theoretical; we also observed this conflict in the ablation study detailed in Section 4.3.

### 3.2.2 PSEUDO-LABEL SCREENING MECHANISM

Corresponding to the calibration heads, the optimization of the cluster head, in turn, relies on the pseudo-labels generated by each calibration head. As mentioned in the introduction, low-quality pseudo-labels hinder the performance of unsupervised clustering and the application of information bottleneck theory. This paper designed a dynamic sample screening mechanism based on information redundancy, thoroughly assessing the learning state of the current calibration head and dynamically selecting high-quality pseudo-labels. This filters out ambiguous samples that the model is "unsure" about, allowing the model to learn from simple and reliable structures first and avoiding premature, overconfident, and incorrect judgments on difficult samples.

Generally, information redundancy is defined as: $R(P) = 1 - \frac{H(P)}{H_{\max}}$ where $H(P) = -\sum_{i=1}^{c} p_i \log_b(p_i)$ and $H_{\max} = H(P_{even})$. Information redundancy can perceive the quality of a probability vector, the more "peaked" the vector, the higher the score. For instance, a one-hot vector receives the highest score, $R(P_{one-hot}) = 1$. The more uniform the vector, the more it represents an inability to distinguish which cluster it belongs to. For example, a $1 \times D$ vector $P_{even} = (\frac{1}{D}, \frac{1}{D}, \ldots, \frac{1}{D})$ receives the lowest score, $R(P_{even}) = 0$. Our screening mechanism is a variant of information redundancy. The quality score for a single sample is:

$$S(P) = 1 - \frac{H(P)}{2 \times H_{\max}} \tag{9}$$

While computing quality scores, samples are ranked according to their top probability values, and the top $K_m^{sel} = \lfloor \sum_{i=1}^{N} S(p_{i,m}) \rfloor$ samples from modality $m$ are selected into the pseudo-label set $S$. As training progresses, the model becomes more confident, quality scores generally increase, and the number of selected samples dynamically grows, allowing the model to learn from more diverse data. This allows the model to learn from simple and reliable structures first, avoiding premature overconfident judgments on difficult samples without negatively impacting the cluster head's performance. See more discussion in Appendix A.2.

### 3.2.3 CLUSTER HEAD

After screening for high-quality pseudo-labels, since the gradient backpropagation from the cluster head is not cut off, we use it to optimize our cluster head and the bottleneck. This allows the calibration signal to be propagated back to the bottleneck, implicitly constructing high-quality target variables for IB in an unsupervised setting. The cluster head is optimized as:

$$\mathcal{L}_{cluH} = -\frac{1}{|S|} \sum_{x_{i,m} \in S} y_{i,m} \log(p_i^{clu}) \tag{10}$$

Here, $y_{i,m}$ is the pseudo-label generated by the $m$-th calibration head for sample $x_i$, and $p_i^{clu}$ is the predicted probability distribution from the cluster head for that sample. By leveraging the pseudo-label set in Equation 10, the calibration signal is delivered to the cluster head. This not only allows the cluster head to perceive modality-specific features but also mitigates the issue of model overconfidence. Concurrently, the calibration signal is transmitted back to IB through the cluster head.

Since the model's core task is multi-modal clustering, and the final result is given by the cluster head which operates on the fused features, the cluster head's objective aligns well with the IB optimization. Therefore allowing gradient backpropagation from the cluster head means that the update signal for the backbones primarily comes from the effectiveness of the fused features. This incentivizes the backbones to learn representations that are "most beneficial for fusion" rather than just optimal for a single modality.

**Theorem 1** (Calibration of biased MI estimation) The pseudo-label screening mechanism allows Equation 10 to selectively learn more or less from the pseudo-labels of a given modality. This enables our model to autonomously rectify the adverse effects caused by modalities with biased MI estimation. See Appendix A.3.

In addition to the aforementioned methods, we also utilize the Kullback-Leibler divergence to enforce consistency among different heads. This regularization term encourages the representations of all views to align within a unified semantic space, enhancing the robustness and discriminative power of the fused representation.

$$\mathcal{L}_{con} = \sum_{m=1}^{M} D_{KL}(p_{:,m}^{cal} \| p^{clu}) \tag{11}$$

Here, $p_{:,m}^{cal}$ represents the clustering result from the $m$-th view. When different modalities produce conflicting clustering results for the same sample, the consistency loss forces the model to output a flatter probability distribution, thereby honestly expressing its uncertainty, effectively lowering ECE and making our framework trustworthy.

### 3.3 OPTIMIZATION

To minimize the first term of $\mathcal{L}_C$, we estimate MI using a variational inference-based approach. As established in **Theorem 2** (See Appendix A.4), this yields a tractable upper bound: $I(X^i; H^i) < \frac{1}{M} \sum_{i=1}^{M} \mathbb{E}_{\theta_i} \{D_{KL}[p(x^i|h^i)||q(x^i)]\}$. Minimizing this bound serves to minimize the MI. To maximize the second term, the computation of MI is thus transformed into the optimization of its tractable lower bound in **Theorem 3** (See Appendix A.5): $I(H^i, H^j) \geq \log(N) - \mathcal{L}_{\text{NT-Xent}}(H^i, H^j)$. This enables us to maximize the lower bound of MI by minimizing the NT-Xent loss. For the optimization of $\mathcal{L}_P$, we use a neural network estimator (Belghazi et al., 2018) to approximate $I(H^m; H^f)$ because $H^m$ and $H^f$ are both high-dimensional variables. This avoids the explicit estimation of probability density for high-dimensional data, which is computationally intractable.

Since $\mathcal{L}_{Cal}$ is directly computable from the equations above, the details of its optimization procedure are omitted here for brevity.

### 3.4 DISCUSSION WITH OTHE METHODS

While CLIB shares methodological similarities with SDCIB(Lou et al., 2025b), MSDIB(Hu et al., 2025), and DDMC(Wang et al., 2025) in their utilization of MI estimation techniques—such as variational inference or neural estimators—it diverges fundamentally in its underlying philosophy and

operational mechanisms. Existing works typically operate under the assumption that these techniques can yield precise MI estimates. However, in the context of high-dimensional and complex data, MI estimation is inherently susceptible to unavoidable estimation bias, thereby compromising the quality of the features extracted by the IB framework. The distinctive advantage of CLIB lies not in the pursuit or assumption of a "flawless" estimator, but in its innovative calibration mechanism. By leveraging complementary cross-modal information, CLIB rectifies the adverse impacts induced by estimation bias, effectively breaking the vicious cycle of "error accumulation" often triggered by noisy pseudo-labels in existing frameworks. It is worth noting that while CLIB is designed to handle biased estimation, the IB framework remains incapable of extracting meaningful features if the estimator fails entirely from the outset. Consequently, we are still motivated to employ the most suitable existing estimation techniques for specific targets to provide the best possible initial feature representation. Furthermore, unlike aforementioned approaches that focus exclusively on clustering accuracy, CLIB is the first to explicitly incorporate ECE into the optimization objective. This integration not only enhances clustering robustness but also effectively mitigates the issue of model overconfidence, providing more reliable decision support for safety-critical applications.

## 4 EXPERIMENT

### 4.1 EXPERIMENTAL SETUP

**Datasets.** We conducted experiments on five widely used benchmark datasets, and a Multi-Layer Perceptron (MLP) was adopted as the backbone for all datasets. Caltech-2V (Fei-Fei et al., 2004): Contains 1,440 images in 7 classes. Each sample is represented by two modalities: Wavelet moments (Shen & Ip, 1999) and CENTRIST features (Wu & Rehg, 2011). Caltech-3V: Identical to Caltech-2V in images and classes, but incorporates a third modality, LBP features (Ojala et al., 2002). ESP-Game (von Ahn & Dabbish, 2005): Comprises 11,032 images across 7 classes, utilizing image features and corresponding textual descriptions as two modalities. IAPR (Grubinger et al., 2006): A collection of images and semantic descriptions. Our study uses a filtered subset of 7,855 images from 6 categories, each with at least four labels. The two modalities are image features and text. MIRFlickr (Huiskes & Lew, 2008): A denoised dataset of 12,154 images in 6 categories, which also employs image features and textual descriptions as its two modalities.

**Implementation Details.** The experiment was conducted with a batch size of 64 using the PyTorch 2.4.1 platform (Python 3.8) on a Windows 10 system equipped with a 24GB NVIDIA RTX-4090D GPU. The cluster head is implemented as an MLP(512d-BN(Ioffe & Szegedy, 2015)-ReLU(Nair & Hinton, 2010)-$C$d), and calibration heads are designed with an identical architecture. We select 4 traditional Single/All modal clustering methods and 11 latest multi-modal clustering methods for comparison to demonstrate the superiority of our method. The classical clustering algorithms are K-Means(KM), Normalized Cuts(Ncuts) and all-view version of them. We run the single-modal method on each modality and report the best clustering results. The rest of selected methods are EAMC (Zhou & Shen, 2020), SiMVC & CoMVC (Trosten et al., 2021), MFLVC (Xu et al., 2022), DSMVC (Tang & Liu, 2022), DealMVC (Yang et al., 2023), ICMVC (Chao et al., 2024), DIVIDE (Lu et al., 2024b), MVCAN (Xu et al., 2024), ROLL (Sun et al., 2025), COPER (Eisenberg et al., 2025).

To ensure that the calibration process is not corrupted by excessive noise, we employ a two-stage training strategy. The first stage is a 100-epoch warm-up period dedicated to stabilizing the feature extraction of IB. Only after the IB can reliably extract features do we proceed to the second stage, a 100-epoch calibration period. We ran the model 20 times, selecting the highest accuracy at the lowest loss to prevent local maxima. For a more comprehensive analysis, the clustering performance was evaluated using two popular metrics: Clustering Accuracy (ACC) (Li & Ding, 2006) and Normalized Mutual Information (NMI) (Strehl & Ghosh, 2002). Higher values of these metrics indicate better clustering performance. The trustworthiness is evaluated using ECE (Zhu et al., 2022). Lower ECE values indicate lower over-confidence and therefore suggest better trustworthiness.

### 4.2 EXPERIMENTAL RESULTS

**Results Analysis.** As shown in Table 1, the proposed CLIB model demonstrates comprehensive superiority, achieving top performance on three evaluation metrics across all five datasets. Re-

Table 1: Clustering performance ACC, NMI (%) and calibration error ECE (%) on five multi-modal datasets (The **bold** denotes the best while underline the second best).

| | Caltech-2V | | | Caltech-3V | | | ESP-Game | | | MIRFlickr | | | IAPR | | |
|---|---|---|---|---|---|---|---|---|---|---|---|---|---|---|---|
| | ACC | NMI | ECE | ACC | NMI | ECE | ACC | NMI | ECE | ACC | NMI | ECE | ACC | NMI | ECE |
| KM | 41.6 | 30.5 | N/A | 46.3 | 31.3 | N/A | 48.4 | 33.5 | N/A | 40.9 | 22.5 | N/A | 38.9 | 17.2 | N/A |
| Ncuts | 39.9 | 31.2 | N/A | 42.6 | 25.4 | N/A | 46.5 | 29.9 | N/A | 48.4 | 26.1 | N/A | 41.9 | 18.9 | N/A |
| ALLKM | 46.4 | 31.4 | N/A | 46.9 | 31.5 | N/A | 34.9 | 20.3 | N/A | 41.0 | 21.6 | N/A | 40.4 | 17.0 | N/A |
| ALLNcuts | 42.8 | 5.2 | N/A | 43.7 | 25.5 | N/A | 33.6 | 18.9 | N/A | 48.2 | 26.2 | N/A | 42.2 | 18.9 | N/A |
| EAMC (CVPR'20) | 40.3 | 26.6 | 35.7 | 38.9 | 21.4 | 19.6 | 27.1 | 6.5 | 24.1 | 30.5 | 9.1 | 29.7 | 37.1 | 16.4 | 24.8 |
| SiMVC (CVPR'21) | 51.1 | 36.9 | 30.2 | 56.9 | 50.4 | 21.3 | 35.3 | 16.2 | 25.6 | 45.6 | 26.3 | 35.4 | 42.7 | 18.5 | 47.9 |
| CoMVC (CVPR'21) | 59.2 | 49.2 | 38.7 | 54.1 | 50.4 | 25.6 | 51.8 | 38.2 | 32.5 | 49.3 | 30.6 | 43.2 | 46.7 | 21.5 | 36.8 |
| DSMVC(CVPR'22) | 57.9 | 49.8 | 31.2 | 65.7 | 54.2 | 26.4 | 32.4 | 27.5 | 31.4 | 48.4 | 29.5 | 31.8 | 38.9 | 15.3 | 24.6 |
| MFLVC (CVPR'22) | 61.5 | 53.6 | 30.3 | 63.1 | 56.6 | 34.2 | 52.1 | **40.1** | 21.7 | 53.8 | 32.8 | 19.5 | 47.3 | 22.6 | 23.6 |
| DealMVC (ACM MM'23) | 47.6 | 37.9 | 19.1 | 59.2 | 56.2 | 27.6 | 42.7 | 24.7 | 23.5 | 49.3 | 32.1 | 21.9 | 35 | 10.8 | 25.8 |
| ICMVC (AAAI'24) | 49.6 | 37.9 | 28.5 | 64.7 | 53.7 | 39.4 | 45.8 | 29.5 | 25.2 | 43.5 | 24.4 | 20.6 | 37.1 | 16.8 | 43.8 |
| DIVIDE (AAAI'24) | 64.1 | 52.9 | N/A | 71.6 | 58.5 | N/A | 46.5 | 27 | N/A | 52.3 | 33.5 | N/A | 45.6 | 23 | N/A |
| MVCAN (CVPR'24) | 46.5 | 37.5 | 23.5 | 70.1 | 62.6 | 24.1 | 50.2 | 35.5 | 33.5 | 50.5 | 31.5 | 26.3 | 34.4 | 17.2 | 22.4 |
| ROLL (CVPR'25) | 45.7 | 35.8 | 19.7 | 61.8 | 49.6 | 16.9 | 41.5 | 23.6 | 25.7 | 46.2 | 27.3 | 24.6 | 38.2 | 17.4 | 23.1 |
| COPER (ICLR'25) | 61.5 | 55.1 | 40.1 | 63.9 | 56.7 | 32.4 | 50.7 | 32.2 | 29.6 | 46.4 | 34.1 | 31.5 | 47.2 | 25.5 | 28.4 |
| CLIB (Ours) | **68.6** | **60.5** | **16.0** | **77.8** | **69.3** | **10.9** | **56.3** | 39.9 | **12.1** | **55.4** | **39.5** | **10.5** | **51.6** | **29.1** | **7.8** |

garding clustering performance, CLIB substantially surpasses previous approaches. For example, it achieves an ACC of 77.8% on the Caltech-3V dataset, outperforming DIVIDE by 6.2%, and an ACC of 56.3% on the ESP-Game dataset, exceeding MFLVC by 4.2%. Similarly, its NMI score of 69.3% on Caltech-3V is significantly higher than the second-best method, MVCAN (62.6%). CLIB demonstrates exceptional performance on the ECE metric, showcasing its strong model calibration capabilities and trustworthiness. On ESP-Game and MIRFlickr, CLIB reduced ECE by more than half compared to the previous best results. Furthermore, on IAPR, its ECE of 7.8% represents a nearly threefold reduction in error compared to the runner-up, MVCAN (22.4%). **Note.** ECE metric is marked as "N/A" for hard-clustering methods like KM and DIVIDE, as ECE is not meaningful for their frameworks.

**Parameter Sensitivity Analysis.** We investigated the parameter sensitivity of our method through a grid search, where hyperparameters $\alpha$ and $\beta$ were varied within the (0,1) interval in 0.1 increments. The experimental results, presented in Figure 2, reveal that our method maintains robust performance across all datasets, with no significant degradation under a wide range of parameter configurations. Specifically, the ACC variance remained within 20% across all datasets, with an average peak-to-trough drop of 17.87%. This observation underscores the method's insensitivity to parameter choices and highlights its reliable and stable nature.

**Convergence Analysis.** Figure 3 illustrates the convergence curves of the overall loss function, ACC, and NMI metrics across all 5 selected datasets. The model's performance improved significantly after the introduction of the calibration module at the 101st epoch. All three metrics stabilized within 40 epochs after this introduction (*i.e.*, at epoch 140), which indicates that our model can converge both quickly and stably.

## 4.3 ABLATION STUDY

To validate the effectiveness of each component and configuration of our framework, we conducted additional ablation experiments. The results in Table 2 indicate that both $\mathcal{L}_{re}$ and $\mathcal{L}_{caliH}$ enhance clustering performance. $\mathcal{L}_{caliH}$ contributes to clustering by filtering high-quality pseudo-labels, which simultaneously increases ACC and reduces ECE. In contrast, while $\mathcal{L}_{con}$ effectively reduces

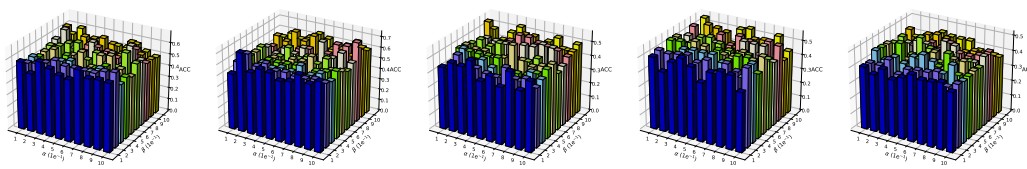

Figure 2: Parameter analysis on Caltech-2V, Caltech-3V, ESP-Game, MIRFlickr and IAPR dataset.

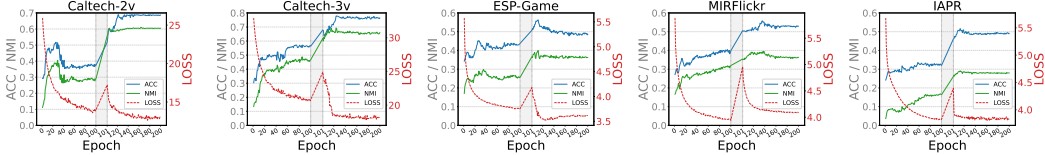

Figure 3: Training process of CLIB on multiple datasets.

Table 2: The ablation results of the proposed model.

| Settings | Caltech-2V | | | Caltech-3V | | | ESP-Game | | | MIRFlickr | | | IAPR | | |
|---|---|---|---|---|---|---|---|---|---|---|---|---|---|---|---|
| | ACC | NMI | ECE | ACC | NMI | ECE | ACC | NMI | ECE | ACC | NMI | ECE | ACC | NMI | ECE |
| $\mathcal{L}_{IB} + \mathcal{L}_{cluH}$ | 57.7 | 43.1 | 36.2 | 60.6 | 52.5 | 25.5 | 43.8 | 31.2 | 21.3 | 51.0 | 34.1 | 25.6 | 39.1 | 18.9 | 35.4 |
| $\mathcal{L}_{IB} + \mathcal{L}_{cluH} + \mathcal{L}_{re}$ | 61.2 | 50.1 | 20.0 | 64.6 | 57.4 | 19.5 | 45.6 | 35.9 | 22.6 | 51.8 | 35.4 | 23.2 | 42.3 | 18.2 | 27.4 |
| $\mathcal{L}_{IB} + \mathcal{L}_{cluH} + \mathcal{L}_{caliH}$ | 61.7 | 53.5 | 19.2 | 67.8 | 59.3 | 11.6 | 46.1 | 35.3 | 21.5 | 53.8 | 36.2 | 18.5 | 39.9 | 20.3 | 18.9 |
| $\mathcal{L}_{IB} + \mathcal{L}_{cluH} + \mathcal{L}_{con}$ | 59.7 | 47.8 | 19.8 | 61.1 | 56.7 | 12.4 | 44.2 | 31.0 | 16.3 | 51.5 | 35.7 | 16.1 | 37.8 | 19.1 | 15.0 |
| I. No warm-up for IB | 59.6 | 57.1 | 19.4 | 61.7 | 58.4 | 15.1 | 48.6 | 34.6 | 21.9 | 52.8 | 38.8 | 27.8 | 40.1 | 22.4 | 32.9 |
| II. Cali heads backprop | 59.9 | 59.3 | 23.4 | 65.9 | 58.8 | 13.5 | 48.5 | 33.4 | 24.3 | 50.4 | 36.2 | 31.3 | 45.4 | 24.4 | 24.3 |
| **CLIB** | **68.6** | **60.5** | **16.0** | **77.8** | **69.3** | **10.9** | **56.3** | **39.9** | **12.1** | **55.4** | **39.5** | **10.5** | **51.6** | **29.1** | **7.8** |

ECE by resolving conflicting views, it yields limited clustering gains. The slight ACC drop on IAPR (39.1% to 37.8%) stems from the constraint preventing the model from making "bold guesses" on ambiguous samples. In safety-critical scenarios where uncertainty is paramount, we suggest increasing the weight of $\mathcal{L}_{con}$ to enhance trustworthiness. Conversely, for applications prioritizing clustering accuracy over calibration, we advise decreasing the weight of $\mathcal{L}_{con}$. The results of the ablation study validate the efficacy of the configurations within our framework, confirming that each component positively contributes to the overall performance of our model.

**I.Remove the warm-up stage for IB.** As shown in Table 2, removing the warm-up stage and applying callibration from the outset introduces substantial noise into the training process. This has two adverse effects: it reduces the quantity of selected pseudo-labels and increases the likelihood of these labels being noisy. Consequently, the model experiences a performance degradation but avoids a complete collapse. This result affirms the necessity of the warm-up stage while simultaneously illustrating the efficacy of the pseudo-label screening mechanism in filtering noise.

**II.Backpropagating the gradient from calibration heads.** Since one calibration head is assigned to each modality, this approach may allow modality-specific redundant information to enter the calibration signal. This information is then passed back to IB via gradient backpropagation, which can impair the bottleneck's ability to learn high-quality features. This ultimately leads to a decline in clustering performance.

## 5 CONCLUSION

This paper introduces CLIB, a novel IB-based multi-modal clustering method. It utilizes a dynamic screening mechanism to learn from high-quality pseudo-labels, which in turn calibrates IB to enhance feature extraction. Supported by theoretical proof and experiments, CLIB achieves competitive clustering performance with lower model overconfidence. Our approach also circumvents the adverse effects of MI estimation bias in high-dimensional data, creating new application possibilities for IB in multi-modal clustering. However, CLIB is currently limited to complete, single-label multi-modal datasets and cannot address incomplete or multi-label data scenarios. Moreover, like many clustering methods shown in Lu et al. (2024a), the reliance on a predefined number of clusters restricts its flexibility. Future research will focus on developing adaptive methods to determine the number of cluster automatically and extending the calibration mechanism to handle missing modalities and multi-label correlations.

## ACKNOWLEDGEMENTS

This work was supported by the National Natural Science Foundation of China (project no. 62576320, 62572444, 62325602), Henan Province Outstanding Youth Science Fund Program (project no. 252300421223).

## ETHICS STATEMENT

This paper does not involve any potential ethics issues.

## REPRODUCIBILITY STATEMENT

We have exerted substantial efforts to guarantee the reproducibility of our research. The main manuscript offers explicit descriptions of the proposed framework, algorithmic modules, and evaluation methodologies. Complete proofs of theoretical findings and detailed derivations of key propositions are included in the appendix. For the experimental section, we utilized public datasets and provided thorough accounts of the experimental setups. To further facilitate reproducibility, the full source code and implementation specifics will be made publicly available upon the paper's acceptance.

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

## A APPENDIX

In the supplemental material:

- Appendix A.1: Statement on the use of large language models.
- Appendix A.2: Discussion about pseudo-label screening mechanism.
- Appendix A.3: Proof of theorem 1.
- Appendix A.4: Proof of theorem 2.
- Appendix A.5: Proof of theorem 3.
- Appendix A.6: T-SNE visualization analysis.
- Appendix A.7: Analysis of modality weights.
- Appendix A.8: Study on difficult samples.
- Appendix A.9: Complexity analysis.

### A.1 STATEMENT ON THE USE OF LARGE LANGUAGE MODELS

In accordance with academic integrity guidelines, we hereby declare that no Large Language Models (LLMs) were used in the conceptualization, experimentation, analysis, or writing of this subsection and the related parts of this paper. All content was produced by the authors independently.

### A.2 DISCUSSION ABOUT PSEUDO-LABEL SCREENING MECHANISM

As discussed in Section 2.2, pseudo-labels generated using fixed thresholds often suffer from low quality and are riddled with noise. To overcome this limitation, researchers began exploring more dynamic pseudo-label refinement strategies. This is evident in the evolution from initial works like CC(Li et al., 2021), which used data augmentation for supervision, to its successor, TCL(Li et al., 2022). TCL introduced a "confidence-based boosting" strategy specifically to dynamically filter reliable pseudo-labels from noisy, self-generated signals, demonstrating that handling unreliable supervision is a critical issue in deep clustering.

For Table 3, CLIB-$S_n$ stands for the version of CLIB applying

$$S_n(P) = 1 - \frac{H(P)}{n \times H_{max}} \tag{12}$$

for quality score function. CLIB-$S_1$ applies the original information redundancy theory and CLIB-$S_2$ is the proposed version. The results indicate that CLIB-$S_2$ is the best-performing version. As the changes become progressively smaller, the performance of CLIB-$S_4$ and subsequent versions ($n \geq 4$) began to converge; therefore, they were excluded from further experimentation and presentation.

Using the original information redundancy theory poses a problem: in an extreme case, if all output vectors from a calibration head are perfectly uniform, then no pseudo-labels from this modality will be selected to guide the cluster head's learning, resulting in a very sparse training signal for the Cluster Head. When a modality is completely unable to distinguish between clusters, but the final output fails to perceive this confusion, the model is highly likely to make overconfident predictions, thereby affecting the clustering result. In contrast, the proposed screening method ensures that at least half of the samples are selected into the pseudo-label set. It is less about selecting high-quality samples but more about excluding the lowest-quality ones.

Jia et al. (2025) and Li et al. (2022) adopted a pseudo-label selection method based on maximum probability. However, by focusing only on the maximum probability, this approach tends to overlook other information provided by the cluster probability vector. As shown in Figure 4, the vector for cluster 2 provides information not only about the most likely cluster but also about the least likely ones. Therefore, it should achieve a higher score from quality score function than the vector for cluster 1 from $S(\cdot)$. If we apply the selection by the maximum probability, they will recieve the same score. We also experimented with a pseudo-label screening mechanism based on maximum probability, which did not perform as well as our proposed method.

Table 3: The results of different screening strategies

| Screening strategies | Caltech-2V | | | Caltech-3V | | | ESP-Game | | | MIRFlickr | | | IAPR | | |
|---|---|---|---|---|---|---|---|---|---|---|---|---|---|---|---|
| | ACC | NMI | ECE | ACC | NMI | ECE | ACC | NMI | ECE | ACC | NMI | ECE | ACC | NMI | ECE |
| Select by max probability | 60.7 | 54.6 | 23.3 | 70.6 | 61.0 | 11.9 | 49.6 | 35.9 | 27.0 | 53.9 | 39 | 35.7 | 42.8 | 26.1 | 21.7 |
| CLIB-$S_1$ | 61,1 | 56.1 | 26.1 | 65.6 | 59.8 | 23.2 | 47.3 | 32.5 | 25.6 | 47.7 | 30.7 | **9.6** | 41.8 | 18.6 | 15.4 |
| **CLIB-$S_2$ (Proposed)** | **68.6** | **60.5** | **16.0** | **77.8** | **69.3** | **10.9** | **56.3** | **39.9** | **12.1** | **55.4** | **39.5** | 10.5 | **51.6** | **29.1** | **7.8** |
| CLIB-$S_3$ | 63.2 | 58.4 | 17.2 | 68.1 | 60.0 | 13.8 | 47.5 | 35.1 | 14.7 | 43.1 | 30.4 | 11.3 | 40.8 | 21.5 | 10.6 |

Figure 4: Different quality score functions for cluster 1 and cluster 2 respectively.

### A.3 PROOF OF THEOREM 1

This proof demonstrates that the proposed calibration mechanism effectively mitigates the adverse effects of estimation biases within IB.

Let $\epsilon_m$ represent the bias in the mutual information estimation for a given modality $m$. The estimated mutual information, $\hat{I}(H^m; H^f)$, is related to the true mutual information, $I(H^m; H^f)$, by:

$$\hat{I}(H^m; H^f) = I(H^m; H^f) + \epsilon_m$$

A large estimation bias, indicated by a large $|\epsilon_m|$, leads IB to produce a less discriminative feature representation $H^m$. This reduction in discriminative power results in the calibration head generating a more uniform, or "flatter," probability distribution $p_m^{cal}$ for the samples of that modality. A flatter distribution is characterized by higher entropy.

Consider two modalities, $m_1$ and $m_2$, with biases $\epsilon_{m_1}$ and $\epsilon_{m_2}$, respectively. If modality $m_1$ has a negligible bias ($\epsilon_{m_1} \to 0$) and modality $m_2$ has a significant bias ($|\epsilon_{m_2}| \gg 0$), the resulting entropy of their respective calibrated probability distributions will satisfy:

$$H(p_{m_2}^{cal}) > H(p_{m_1}^{cal})$$

The sample quality score, $S(P)$, is designed to be negatively correlated with the entropy $H(P)$, as shown by its derivative:

$$\frac{\partial S(P)}{\partial H(P)} = -\frac{1}{2H_{max}} < 0$$

The number of high-quality samples selected for modality $m$, denoted $K_m^{sel}$, is calculated as $K_m^{sel} = \left\lfloor \sum_{i=1}^{N} S(p_{i,m}^{cal}) \right\rfloor$. This establishes a direct link between estimation bias and the number of selected samples. The logical chain is as follows:

$$H(p_{m_2}^{cal}) > H(p_{m_1}^{cal}) \implies S(p_{m_2}^{cal}) < S(p_{m_1}^{cal}) \implies K_{m_2}^{sel} < K_{m_1}^{sel}$$

In summary, a modality with a larger estimation bias will have fewer samples selected for the subsequent training step.

We now analyze how this sample selection mechanism influences the parameter updates of IB, denoted by $\theta$. The objective is to minimize the cluster head loss, $\mathcal{L}_{cluH}$, which is computed using the union of selected samples from all modalities. The total gradient of the loss with respect to the IB parameters $\theta$ is the sum of contributions from each modality:

$$\nabla_\theta \mathcal{L}_{cluH} = \sum_{m=1}^{M} G_m$$

where $G_m$ is the gradient contribution originating from the set of samples $S_m$ selected for modality $m$, with $|S_m| = K_m^{sel}$. From Equation 10, this contribution is defined as the gradient of the loss computed exclusively on these selected samples:

$$G_m = \nabla_\theta \mathcal{L}_{cluH}^{(m)} = \nabla_\theta \left( \sum_{x_i \in S_m} -y_{i,m} \log(p_i^{clu}) \right)$$

By linearity of the gradient operator, this can be expressed as:

$$G_m = \sum_{x_i \in S_m} \nabla_\theta \left( -y_{i,m} \log(p_i^{clu}) \right)$$

The magnitude of this gradient contribution, $\|G_m\|$, is determined by the vector sum of the individual sample gradients. We assume that for a given modality, the gradient vectors for each sample are directionally coherent, as they all contribute to optimizing the same feature extraction parameters $\theta$. Under this reasonable assumption, the magnitude of the summed vector is approximately proportional to the number of terms in the summation:

$$\|G_m\| = \left\| \sum_{x_i \in S_m} \nabla_\theta \left( -y_{i,m} \log(p_i^{clu}) \right) \right\| \propto |S_m| = K_m^{sel}$$

Combining the findings from both sections, we see that for our two modalities $m_1$ (low bias) and $m_2$ (high bias), we have $K_{m_1}^{sel} > K_{m_2}^{sel}$. This directly implies that the magnitude of their respective gradient contributions will satisfy:

$$\|G_{m_1}\| > \|G_{m_2}\|$$

During optimization, the total gradient $\nabla_\theta \mathcal{L}_{cluH} = G_1 + G_2 + \cdots + G_M$ is a vector sum. In this sum, the gradient vector $G_1$, originating from the more reliable modality with a smaller estimation bias, possesses a significantly greater magnitude. Consequently, the direction of the total gradient update will be predominantly influenced by $G_1$. This ensures that IB's parameters are primarily updated based on information from more reliable modalities, effectively mitigating the negative impact of modalities corrupted by high estimation bias.

### A.4 PROOF OF THEOREM 2

In our model, a key optimization objective is to minimize the mutual information $I(X^i; H^i)$ between the input data $X^i$ and its compact representation $H^i$. However, the direct computation of this mutual information is intractable as it involves the posterior probability $p(x^i|h^i)$, which is difficult to handle. Similar to Hu et al. (2025), we employ a variational inference approach to derive a trainable approximate upper bound, and by lowering it, we could minimize $I(X^i; H^i)$.

First, the mutual information $I(X^i; H^i)$ is defined as:

$$I(X^i; H^i) = \int_{h^i} \int_{x^i} p(x^i, h^i) \log \frac{p(x^i|h^i)}{p(x^i)}.$$

Since the posterior probability $p(x^i|h^i)$ is unknown, we introduce a parameterized variational distribution $q(x^i)$ to approximate the true marginal probability $p(x^i)$. To ensure the effectiveness of this approximation, we utilize the Kullback-Leibler (KL) divergence to constrain the distance between $q(x^i)$ and $p(x^i)$. Based on the non-negativity of KL divergence:

$$D_{KL}(p(x^i)||q(x^i)) = \int p(x^i) \log \frac{p(x^i)}{q(x^i)} \geq 0.$$

Through a simple transformation, we can obtain:

$$\int p(x^i) \log p(x^i) - \int p(x^i) \log q(x^i) \geq 0$$

$$\Rightarrow p(x^i) \geq q(x^i).$$

This implies that substituting $p(x^i)$ with $q(x^i)$ provides an upper bound for the original mutual information expression:

$$I(X^i; H^i) = \int \int p(x^i, h^i) \log \frac{p(x^i|h^i)}{p(x^i)} \leq \int \int p(x^i, h^i) \log \frac{p(x^i|h^i)}{q(x^i)}.$$

Next, by leveraging the property of the joint probability distribution, $p(x^i, h^i) = p(h^i)p(x^i|h^i)$, we can rewrite the inequality as:

$$I(X^i; H^i) \leq \int \int p(h^i)p(x^i|h^i) \log \frac{p(x^i|h^i)}{q(x^i)}.$$

Table 4: Variable definitions of Theorem 3

| Variable | Definition |
|---|---|
| $H_u, H_v$ | A batch of feature representations from modalities $u$ and $v$. |
| $N$ | The batch size. |
| $h_i^u \in H_u$ | The feature vector of the $i$-th sample from modality $u$. |
| $h_i^v \in H_v$ | The feature vector of the $i$-th sample from modality $v$. |
| $(h_i^u, h_i^v)$ | A "positive pair," sampled from the joint distribution $p(h^u, h^v)$. |
| $sim(h_1, h_2)$ | A similarity metric for feature vectors, typically cosine similarity: $sim(h_1, h_2) = \frac{h_1^T h_2}{\|h_1\|\|h_2\|}$. |
| $\tau$ | A temperature hyperparameter. |

To eliminate extraneous elements, we use Monte Carlo sampling (Hastings, 1970) to approximate the integral over $h^i$, which yields:

$$I(X^i; H^i) \leq \int p(x^i|h^i) \log \frac{p(x^i|h^i)}{q(x^i)}.$$

We assume that $p(x^i|h^i)$ follows a Gaussian distribution, with its mean and variance learned by the sharing specific encoder network (Alemi et al., 2017; Mao et al., 2021). Here we use the reparameterization trick, expressing $h^i$ as:

$$h^i = \mu(x^i) + \sigma(x^i) \cdot \theta, \quad \text{where} \quad \theta \sim \mathcal{N}(0, I).$$

The upper bound of the mutual information can thus be expressed as an expectation with respect to $\theta$, and is ultimately simplified to the expectation of a KL divergence:

$$I(X^i; H^i) \leq \mathbb{E}_{\theta_i} \left[ \log \frac{p(x^i|h^i)}{q(x^i)} \right] \leq \mathbb{E}_{\theta_i} \{ D_{KL}[p(x^i|h^i)||q(x^i)] \}.$$

To prevent the model from producing a trivial solution by assigning all samples to a few clusters, we impose a constraint on $q(x_i)$ to enforce a uniform distribution over the classes: $\sum_i^N q(x_i) = \frac{N}{|C|}$, where $N$ is the total number of samples and $|C|$ is the number of clusters. Finally, by averaging over all samples in the dataset, we obtain the final optimizable objective function:

$$I(X^i; H^i) < \frac{1}{N} \sum_{i=1}^{N} \mathbb{E}_{\theta_i} \{ D_{KL}[p(x^i|h^i)||q(x^i)] \}.$$

## A.5 Proof of Theorem 3

The core idea of this proof is that the NT-Xent loss function is, in fact, an estimator for a lower bound on the mutual information between the representations of two modalities. Consequently, by minimizing the NT-Xent loss, we are effectively maximizing this lower bound, which serves as a proxy for estimating the mutual information itself.

The standard definition of the mutual information can be expressed as the KL divergence between the joint distribution $p(h^u, h^v)$ and the product of the marginal distributions $p(h^u)p(h^v)$. In expectation form, this is:

$$I(H_u, H_v) = \mathbb{E}_{p(h^u, h^v)} \left[ \log \frac{p(h^u, h^v)}{p(h^u)p(h^v)} \right] = \mathbb{E}_{p(h^u, h^v)} \left[ \log \frac{p(h^v|h^u)}{p(h^v)} \right]$$

Directly computing this MI is intractable, as it requires knowledge of the high-dimensional probability density functions $p(h^v|h^u)$ and $p(h^v)$. The Noise-Contrastive Estimation (InfoNCE) framework provides a way to estimate a lower bound on this quantity.

The core idea of InfoNCE is to introduce a scoring function (or critic), $f(h^u, h^v)$, designed to approximate the log-density ratio:

$$f(h^u, h^v) \approx \log \frac{p(h^v|h^u)}{p(h^v)}$$

We can formalize this by stating that the density ratio is proportional to the exponentiated score function:

$$\frac{p(h^v|h^u)}{p(h^v)} \propto \exp(f(h^u, h^v))$$

To convert this proportionality into an equation, we must normalize the expression. The normalization term is found by ensuring that the probability distribution integrates to one, which is achieved by taking the expectation over all possible $h^v$ drawn from the marginal distribution $p(h^v)$:

$$\frac{p(h^v|h^u)}{p(h^v)} = \frac{\exp(f(h^u, h^v))}{\mathbb{E}_{h'^v \sim p(h^v)}[\exp(f(h^u, h'^v))]}$$

Substituting this back into the definition of mutual information yields an expression based on our critic $f$:

$$I(H_u, H_v) = \mathbb{E}_{p(h^u, h^v)}\left[f(h^u, h^v) - \log\left(\mathbb{E}_{h'^v \sim p(h^v)}[\exp(f(h^u, h'^v))]\right)\right]$$

The expectation term $\mathbb{E}_{h'^v \sim p(h^v)}[\cdot]$ in the denominator is still intractable. We can, however, approximate it using a Monte Carlo method. We draw $N$ independent and identically distributed samples $\{h_j^v\}_{j=1}^N$ from the marginal distribution $p(h^v)$. For a sufficiently large $N$, the Law of Large Numbers states:

$$\mathbb{E}_{h'^v \sim p(h^v)}[\exp(f(h^u, h'^v))] \approx \frac{1}{N}\sum_{j=1}^N \exp(f(h^u, h_j^v))$$

By substituting this Monte Carlo approximation, we introduce an inequality. Because the logarithm $\log(\cdot)$ is a concave function, Jensen's Inequality ($\mathbb{E}[\log(X)] \leq \log(\mathbb{E}[X])$) tells us that replacing the log of the true expectation with the log of the sample mean results in a lower bound. Therefore:

$$I(H_u, H_v) \geq \mathbb{E}_{(h_i^u, h_i^v) \sim p(h^u, h^v)}\left[f(h_i^u, h_i^v) - \log\left(\frac{1}{N}\sum_{j=1}^N \exp(f(h_i^u, h_j^v))\right)\right]$$

This expression is the InfoNCE lower bound on mutual information.

The NT-Xent loss is a specific instance of the InfoNCE objective. In this context, the scoring function $f(h_1, h_2)$ is defined as the temperature-scaled cosine similarity:

$$f(h_1, h_2) = \text{sim}(h_1, h_2)/\tau$$

The NT-Xent loss for a batch is defined as:

$$\mathcal{L}_{\text{NT-Xent}}(H_u, H_v) = -\frac{1}{N}\sum_{i=1}^N \log \frac{\exp(\text{sim}(h_i^u, h_i^v)/\tau)}{\sum_{j=1}^N \exp(\text{sim}(h_i^u, h_j^v)/\tau)}$$

We can now show the direct relationship between the InfoNCE bound and this loss function. In a practical implementation, the expectation over positive pairs, $\mathbb{E}_{(h_i^u, h_i^v)}$, is approximated by the empirical mean over the $N$ pairs in the batch. Starting with the InfoNCE lower bound:

$$I(H_u, H_v) \geq \frac{1}{N}\sum_{i=1}^N\left[f(h_i^u, h_i^v) - \log\left(\frac{1}{N}\sum_{j=1}^N \exp(f(h_i^u, h_j^v))\right)\right]$$

$$= \frac{1}{N}\sum_{i=1}^N\left[\log(\exp(f(h_i^u, h_i^v))) - \log\left(\frac{1}{N}\sum_{j=1}^N \exp(f(h_i^u, h_j^v))\right)\right]$$

$$= \frac{1}{N}\sum_{i=1}^N \log \frac{\exp(f(h_i^u, h_i^v))}{\frac{1}{N}\sum_{j=1}^N \exp(f(h_i^u, h_j^v))}$$

$$= \frac{1}{N}\sum_{i=1}^N\left[\log \frac{\exp(f(h_i^u, h_i^v))}{\sum_{j=1}^N \exp(f(h_i^u, h_j^v))} + \log(N)\right]$$

$$= \left(\frac{1}{N}\sum_{i=1}^N \log \frac{\exp(f(h_i^u, h_i^v))}{\sum_{j=1}^N \exp(f(h_i^u, h_j^v))}\right) + \left(\frac{1}{N}\sum_{i=1}^N \log(N)\right)$$

$$= -\mathcal{L}_{\text{NT-Xent}}(H_u, H_v) + \log(N)$$

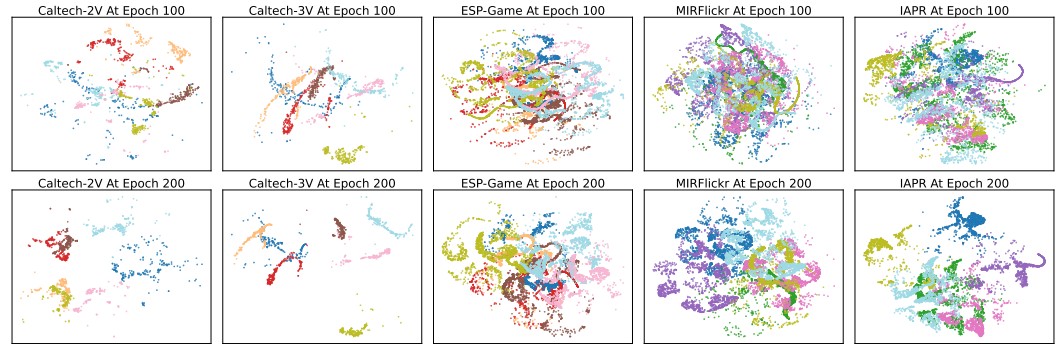

Figure 5: T-sne visualization results on multiple datasets.

This final expression, $I(H_u, H_v) \geq \log(N) - \mathcal{L}_{\text{NT-Xent}}(H_u, H_v)$, demonstrates the connection. For a fixed batch size $N$, a smaller $N$ may result in a looser lower bound for the MI estimation. However, since $\log(N)$ is a constant, its derivative with respect to the model parameters is zero. Therefore, this term does not influence the direction of the gradient update in Equation 3. From the above, minimizing the NT-Xent loss is equivalent to maximizing a variational lower bound on the mutual information $I(H_u, H_v)$.

### A.6 T-SNE VISUALIZATION ANALYSIS

To further illustrate the impact of the calibration mechanism on the final clustering structure, Figure 5 presents t-SNE visualizations for all datasets at epoch 100 (prior to the introduction of calibration) and epoch 200 (after calibration). It can be observed that following 100 epochs of calibration, the clustering structures across all datasets become significantly more compact and well-separated. On the Caltech-3V dataset, where the model achieves its best performance, the clusters are clearly distinct; data samples within most clusters are densely distributed, while samples from different clusters maintain a substantial separation distance.

### A.7 ANALYSIS OF MODALITY WEIGHTS

To investigate the dynamic adjustment capability of the CLIB framework during multi-modal fusion, we tracked the evolution of modality weights ($w_m$) throughout the training process across five datasets. As illustrated in Figure 6, the weight variations in all datasets exhibit a distinct two-stage characteristic, which aligns consistently with our designed training strategy. In the MIRFlickr, ESP-Game, and IAPR datasets, the weights diverged rapidly, with the model tending to assign excessively high weights to a dominant modality. However, upon the introduction of the Calibration Mechanism at epoch 100, significant turning points appeared in the weight curves of all datasets, verifying the effectiveness of the calibration signals. Following this shift, the weights stabilized quickly. This demonstrates that CLIB's adaptive weighted-average fusion mechanism is capable of dynamically allocating weights based on the actual discriminability of each modality.

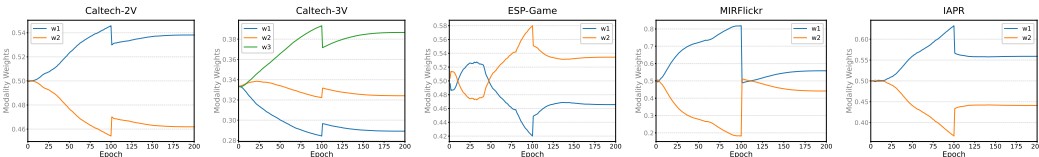

Figure 6: Evolution of modality fusion weights on multiple datasets

Table 5: Clustering performance on difficult samples.

| | Caltech-2V | | | Caltech-3V | | | ESP-Game | | | MIRFlickr | | | IAPR | | |
|---|---|---|---|---|---|---|---|---|---|---|---|---|---|---|---|
| | ACC | NMI | ECE | ACC | NMI | ECE | ACC | NMI | ECE | ACC | NMI | ECE | ACC | NMI | ECE |
| At epoch 101 | 28.6 | 27.2 | 28.2 | 18.9 | 24.4 | 25.5 | 27.2 | 15.2 | 21.9 | 27.9 | 17.2 | 27.4 | 26.8 | 14.6 | 32.9 |
| At epoch 200 | 43.1 | 39.5 | 21.4 | 46.7 | 28.3 | 19.6 | 35.2 | 19.7 | 19.8 | 30.5 | 21.6 | 21.5 | 39.7 | 17.9 | 24.3 |

## A.8 STUDY ON DIFFICULT SAMPLES

During the initial stage of calibration, samples with low quality score were excluded from the pseudo-label set to filter out noise. To investigate the final clustering performance of these samples, we defined the bottom 20% of samples—ranked by their quality score at the 101st epoch—as "difficult samples." Their final performance is presented in Table 5. We recorded the clustering performance at epochs 101 and 200. The results indicate that after 100 epochs of calibration, the performance of hard samples improved. This is attributed to the fact that, although hard samples were initially excluded from the pseudo-label set, the quality scores for all samples gradually increased as the model learned. Consequently, an increasing number of samples entered the pseudo-label set, enabling the model to eventually learn from these hard samples.

## A.9 COMPLEXITY ANALYSIS

Let $D$ denote the input dimension of the samples, $L_B$ and $P_B$ represent the number of layers and the hidden dimension of the backbone network, respectively, and $H_D$ be the dimension of the features output by the backbone. The $I(X^i, H^i)$ term has a complexity of $\mathcal{T}_{Backbone} \approx O(N \cdot M \cdot (DP_B + (L_B - 1)P_B^2 + P_B H_D))$. Concurrently, the $I(H^i, H^j)$ term's total complexity is $O(M^2 N^2 H_D)$. Finally, the $I(H^m, H^f)$ term, calculated via a neural network estimator, incurs a cost of $O(NMP_{MINE})$, assuming a cost of $O(NP_{MINE})$ per modality. Therefore, the total complexity for $\mathcal{L}_{IB}$ is $\mathcal{T}_{IB} = \mathcal{T}_{Backbone} + O(M^2 N^2 H_D) + O(NMP_{MINE})$.

The computational cost of $\mathcal{L}_{Cal}$ is determined by three primary operations. First, the $M + 1$ heads, which map $H_D$-dimensional inputs to $C$ outputs via a $P_H$-dimensional hidden layer, introduce a complexity of $\mathcal{T}_{Heads} = O((M+1) \cdot N \cdot (H_D P_H + P_H C))$. Second, the $\mathcal{L}_{caliH}$ calculation requires a preceding K-Means step, running $K_M$ iterations for $C$ clusters on $N$ $H_D$-dimensional features, which incurs a cost of $O(K_M NCH_D)$. Third, the pseudo-label screening for $N \cdot M$ samples requires approximately $O(NMC)$. The remaining losses are computed in $O(NMC)$ or less. Thus, the total complexity for the $\mathcal{L}_{Cal}$ module is $\mathcal{T}_{Cal} \approx \mathcal{T}_{Heads} + O(K_M NCH_D) + O(NMC)$.

Let $K_{warmup}$ be the number of warm-up epochs and $K_{cal}$ be the number of calibration epochs. The total time complexity $\mathcal{T}_{total} = K_{warmup} \cdot \mathcal{T}_{IB} + K_{cal} \cdot (\mathcal{T}_{IB} + \mathcal{T}_{Cal})$.

