# OpenReview forum: "Calibrated Information Bottleneck for Trusted Multi-modal Clustering"
_ICLR.cc/2026/Conference — ICLR 2026 Poster_

### Official Review · Reviewer_omRQ · 2025-10-28

**Soundness:** 3
**Presentation:** 3
**Contribution:** 3
**Rating:** 6
**Confidence:** 4

**Summary:**

This paper proposes CLIB, a calibrated information bottleneck framework for multi-modal clustering. It introduces a multi-head architecture with dedicated calibration heads to mitigate the impact of noisy pseudo-labels and biased mutual information estimation—key issues in existing IB-based methods. A dynamic label selection strategy further improves training stability. Experiments show CLIB achieves state-of-the-art clustering accuracy and superior calibration on multiple benchmarks.

**Strengths:**

-Well-motivated problem: The focus on improving trustworthiness and calibration in multi-modal clustering is timely and important.
-Innovative architecture: The multi-head design with dedicated calibration heads is a clever way to decouple representation learning from label refinement.

**Weaknesses:**

-Limited discussion on calibration in clustering: While ECE is adapted for clustering, the paper does not fully address the challenge of defining "correctness" without ground-truth labels during calibration evaluation.
-Lack of computational analysis: No comparison of training time or model complexity is provided, making it hard to assess practical efficiency.
-Hyperparameter sensitivity: The balance between IB objectives and calibration is controlled by hyperparameters; their robustness is not thoroughly analyzed.

**Questions:**

The method relies on pseudo-labels for both clustering and calibration, yet these labels are inherently noisy and evolve during training. How does the proposed calibration mechanism avoid reinforcing or amplifying incorrect pseudo-labels in the early or unstable stages of training? A brief analysis or design justification on the robustness of calibration to label noise would significantly strengthen the paper's claim of producing "trusted" clustering.

---

> ### Author Response · Authors · 2025-11-20
> **Response to Reviewer omRQ**
>
> Thank you for this thoughtful comment.
> ***
> **Weakness1**
> \
> \
> In training process of unsupervised clustering, the unavailability of ground-truth labels necessitates an alternative standard for validation. To address this, CLIB implicitly defines “correctness” by assessing the trustworthiness of self-generated pseudo-labels via a dedicated quality scoring mechanism. During training, the quality function $S(\cdot)$ (**Eq. 9**) serves as a quantifiable proxy for correctness, where samples achieving higher scores—indicative of low uncertainty and distinct cluster assignment—are deemed 'correct' and selected to guide the learning process, effectively filtering out ambiguous data that might otherwise propagate noise.
>
> However, for model evaluation, we employ ground-truth labels, which is a standard practice in the field of clustering algorithms. Without these labels, it is impossible to objectively measure whether the algorithm has correctly uncovered the latent semantic structure of the data. For instance, Clustering Accuracy (ACC), one of the most widely used metrics, necessitates ground-truth labels for its calculation. Regarding trustworthiness, while we could employ internal metrics—such as the quality function $S(\cdot)$ defined in this work—to gauge performance, such approaches are inherently subjective. A model may be biased without being aware of it. Consequently, we have excluded this internal indicator from the experimental section. To objectively assess the trustworthiness of the model, the use of Expected Calibration Error (ECE), calculated using ground-truth labels, remains indispensable.
>
> **Revision:**
>
> We have revised Section 1 (Introduction) to emphasize the use of pseudo-labels as a proxy for correctness.
>
> ***
> **Weakness2**
> \
> \
> We appreciate the reminder from you and Reviewer VCs7. we have recognized the need to analyze the model's computational complexity.
>
> **Revision:**
>
> In the revised manuscript, we have added Section 3.4 Complexity Analysis, which is dedicated to discussing computational complexity. Please refer to the revised manuscript for the detailed discussion.
>
> ***
> **Weakness3**
> \
> \
> Parameter sensitivity analysis shows that the ACC variance remained within 20% across all datasets, with an average peak-to-trough drop of 17.87%. This confirms that the method suffers no significant performance degradation.
>
> **Revision:**
>
> We have added a more detailed discussion on parameter sensitivity in **Section 4.2 (Experimental Results)**.
> ***
> **Question**
> \
> \
> We have adopted the following strategies to avoid the impact of noisy labels:
>
> 1.We introduced a pseudo-label selection mechanism to perceive the clustering quality of individual modalities and subsequently filter for high-confidence and low-noise pseudo-labels to supervise the model's training.
>
> 2.We adopt a two-stage training methodology that warms up the IB to significantly reduce the noise propagated into the calibration stage.
>
> 3.For samples where modalities yield conflicting opinions, the framework forces the model to output a flatter probability distribution, thereby ensuring the model avoids overconfidence. Furthermore, this flatter distribution results in a lower score from the quality function $S(\cdot)$, which subsequently reduces the likelihood of the sample being selected for the pseudo-label set, effectively preventing the influence of noisy labels.
>
> These designs are detailed in **Section 3.1 (Representation Learning Based on Information Bottleneck)**, **Section 3.2.2 (Pseudo-label Screening Mechanism)**, and **Section 3.2.3 (Cluster Head)**, respectively, and are further analyzed in **Section 1 (Introduction)**.
>
> However, we do not permanently exclude low-quality labels. As the model's learning progresses, these noisy, low-quality samples—namely, difficult samples—will eventually enter the pseudo-label set to be learned. Furthermore, as the calibration proceeds, their clustering performance improves. This is demonstrated in our newly added **Section 4.4 (Study on Difficult Samples)**.
>
> We hope this clarification addresses your concerns.
>
>
> **Revisions:**
>
> 1.In the analysis within **Section 1 (Introduction)**, we have further emphasized the framework's capability to avoid noise.
>
> 2.We added **Section 4.4 (Study on Difficult Samples)** to show that the clustering performance of noisy “difficult samples” is also improved via the calibration mechanism.
>
> ***
>
> The above revisions have been marked in blue in the revised version.
>
> If your concerns have been addressed, Could you please help raise the score. If you have any other concerns, please let us know, and we will try our best to address them. Thanks.

---

> > ### Comment · Reviewer_omRQ · 2025-11-26
> >
> > The author has addressed all my concerns, so I believe the original decision should stand.

---

> > > ### Author Response · Authors · 2025-11-26
> > >
> > > Dear Reviewer omRQ,
> > > \
> > > \
> > > Thank you for acknowledging the revisions and confirming that the concerns have been addressed. I truly appreciate your time and constructive feedback throughout this process.
> > >
> > > If applicable, could I kindly request your consideration of raising the rating to reflect the revised manuscript's improvements?  This would greatly help the AC/SAC in their final decision.
> > >
> > > Please let me know if any further clarifications are needed.
> > > \
> > > Best regards
> > >
> > > Authors of paper 4840

---

> > > ### Author Response · Authors · 2025-11-27
> > >
> > > Dear Reviewer  omRQ,
> > > \
> > > \
> > > Thank you for your time in reviewing the revised manuscript and for confirming that the revisions addressed your concerns. I truly appreciate your constructive feedback, which has significantly strengthened the paper.
> > >
> > > As the current version now incorporates all suggestions from reviewers, I hope these improvements align with your expectations.
> > >
> > > Given that three other reviewers have agreed to rate the paper as 8/10 after the rebuttal, would you kindly consider updating your rating to reflect the revised manuscript's quality? This would greatly assist the AC/SAC in reaching a fair final decision.
> > >
> > > Please let me know if any additional information would be helpful.
> > >
> > > \
> > > \
> > > Best regards,
> > >
> > > Authors of paper 4840

---

### Official Review · Reviewer_ofuD · 2025-10-30

**Soundness:** 3
**Presentation:** 3
**Contribution:** 3
**Rating:** 6
**Confidence:** 5

**Summary:**

Information bottleneck theory effectively removes redundancy or noise in multi-modal clustering while preserving discriminative information, but existing IB-based methods face challenges such as low-quality pseudo-labels, over-reliance on accurate mutual information estimation and unreliable clustering outcomes. This work proposes a calibrated information bottleneck framework featured a parallel multi-head network to reach three typical goals, calibrating biased MI estimation to enhance IB stability, building reliable IB targets from multi-modal data samples and getting trusted results. A dynamic pseudo-label selection strategy grounded in information redundancy theory improves training stability by filtering high-confidence labels. Experiments show the method achieves promising results across many benchmarks.

**Strengths:**

1.The proposed CLIB framework is innovative. Its parallel multi-head architecture successfully decouples the calibration objective from the final clustering objective.

2.Theorem 1 given in the paper is insightful. It theoretically connects the difficult problem of MI estimation bias with the pseudo-label screening mechanism via a clear logical chain.

3.The experimental section is solid. On five widely-used benchmark datasets, CLIB achieves state-of-the-art performance on all three metrics. Particularly, the significant reduction in ECE robustly demonstrates CLIB's effectiveness in mitigating model overconfidence.

**Weaknesses:**

1.One of the core motivations of the paper is that MI estimation is difficult and biased. However, in the actual implementation, different parts of the model use three different MI estimation strategies. The authors do not explicitly explain the rationale for choosing this.

2.The implementation environments of the code are not given in the experiments, which may influence the reproductivity. A complete experimental details is usually needed in the experimental setup subsection.

**Questions:**

1. In the adaptive fusion, could you provide the specific weights w_m learned during the experiments? This would help verify whether the model is truly adapting the importance of different modalities.

2. The ablation study for L_con shows a potential trade-off between ACC and ECE.  In a practical application, if one must prioritize one over the other, how would you advise them to adjust the model?

3. Is this the first work addressing the trusted multi-modal clustering problem with calibration? If yes, please explicitly describe it in the paper. If not, please deeply discuss the differences with the related works.

---

> ### Author Response · Authors · 2025-11-20
> **Response to Reviewer ofuD**
>
> We greatly appreciate the reviewer’s thoughtful feedback and recognition of the contributions of our work.
> ***
> **Weakness1**
> \
> \
> While CLIB can handle biased mutual information estimation, if the estimator fails completely from the outset, the IB will be unable to learn any meaningful features. Therefore, we are still motivated to use the most suitable existing estimation techniques for each specific target to provide the best possible initial feature representation.
>
>
>
> 1. To achieve effective information compression, we aim to minimize the mutual information $I(X^{i};H^{i})$. Since this optimization is equivalent to minimizing a tractable upper bound, we employ a variational inference-based approach by introducing a variational distribution to approximate the true posterior.
> 2. To achieve feature alignment, we need to maximize the mutual information $I(H^i, H^j)$ between representations of different modalities. As this requires maximizing a lower bound, we utilize the NT-Xent loss, which inherently optimizes the InfoNCE lower bound.
> 3. For the maximization of $I(H^m; H^f)$, where explicit probability density estimation is intractable due to high-dimensional variables, we employ a neural network estimator to approximate it.
>
> ***Revision:***
>
> In **Section 3.3 (Optimization)** of the revised manuscript, we briefly justified our choice of the corresponding mutual information estimation methods.
> ***
> **Weakness2**
> \
> \
> We appreciate your reminder to include the implementation environments of the experiment. The experiment was conducted using the PyTorch 2.4.1 platform (Python 3.8) on a Windows 10 system equipped with a 24GB NVIDIA RTX-4090D GPU. We hope this clarification addresses your concerns.
>
> ***Revision:***
>
> Following your suggestion, we have added the implementation environments to **Section 4.1 (Experimental Setup)** in the revised manuscript.
> ***
> **Question1**
> \
> \
> **Figure 6** in the revised manuscript illustrates a two-stage weight evolution consistent with our strategy. Initial divergences toward dominant modalities were effectively corrected upon introducing the calibration mechanism at epoch 100. The subsequent rapid stabilization confirms that CLIB dynamically and accurately allocates weights according to the discriminability of each modality.
>
> To provide a more intuitive representation of the final modality weights learned by CLIB, we list the converged weights in the table below.
>
>
> ###  The weights of each modality after convergence on various datasets.
> |    | Caltech-2V | Caltech-3V | ESP-Game | MIRFlickr | IAPR |
> |:---|:---:|---:|:---:|:---:|:---:|
> | W1 | 0.5381 | 0.2892 | 0.4657 | 0.5584 | 0.559 |
> | W2 | 0.4619 | 0.3242 | 0.5343 | 0.4416 | 0.441 |
> | W3 | \ | 0.3867 | \ | \ | \ |
>
>
> ***Revision:***
>
> we have included **Appendix A.7 (Analysis of Modality Weights)** and **Figure 6** in the revised manuscript. This appendix elaborates on the dynamic adjustment capability of the CLIB framework during multi-modal fusion, and demonstrates the impact of introducing the calibration mechanism on weight learning.
>
> ***
> **Question2**
> \
> \
> In the ablation study, adding $\mathcal{L}\_{con}$ caused a slight ACC decrease (39.1% to 37.8%) on the IAPR dataset.
> This phenomenon occurs because $\mathcal{L}\_{con}$ prioritizes trustworthiness over discriminative power in ambiguous cases. When the model encounters conflicting information from different modals, $\mathcal{L}\_{con}$ compels the model to smooth its prediction distribution to lower the ECE. While this effectively mitigates overconfidence and reduces ECE, it prevents the model from making "bold guesses" on difficult samples, leading to a slight sacrifice in ACC in exchange for higher reliability.
> In safety-critical scenarios where knowing model’s uncertainty is paramount, we suggest increasing the weight assigned to $\mathcal{L}\_{con}$ to enhance trustworthiness. On the other hand, for applications that prioritize clustering accuracy over calibration and can tolerate a degree of overconfidence, we advise decreasing the weight of $\mathcal{L}\_{con}$.
>
> ***Revision:***
>
> We have added an explanation in **Section 4.2 (Ablation Study)**.
> ***
> **Question3**
> \
> \
> Indeed, to the best of our knowledge, this is the first work to address the trusted multi-modal clustering problem using calibration. We agree that explicitly claiming this work as the first of its kind in the domain is essential. We appreciate you bringing this to our attention.
>
> ***Revision:***
>
> We have clarified this point in **Section 1 (Introduction)** of the revised manuscript.

---

> > ### Comment · Reviewer_ofuD · 2025-11-27
> >
> > Thanks for addressing my questions in a detailed manner. Thus I tend to raise my rating to 8 and fully agree to accept this work.

---

### Official Review · Reviewer_VCs7 · 2025-10-31

**Soundness:** 3
**Presentation:** 3
**Contribution:** 3
**Rating:** 6
**Confidence:** 5

**Summary:**

This paper introduces an information bottleneck-based multi-modal clustering method, which addresses the issue of overconfident clustering results caused by low-quality or unreliable pseudo-labels in multi-modal clustering. The parallel multi-head architecture is proposed to correct distortions in mutual information estimation. Experiments demonstrate that the proposed method achieves superior performance compared with existing baselines across multiple datasets.

**Strengths:**

1. This work addresses the reliance on reliable target variables and the overconfidence caused by noisy pseudo-labels, which is a commonly encountered problem in clustering.
2. The proposed dynamic pseudo-label screening strategy based on information redundancy offers a promising alternative to existing probability-based thresholding.
3. The proposed framework takes the inherent difficulty of precise MI estimation into consideration, alleviating the negative impacts of such estimation biases.

**Weaknesses:**

1. The proposed method consists of M+1 heads, a two-stage training process, and multiple loss terms, which may increase computational overhead compared to baseline methods. The authors are encouraged to discuss the model’s computational complexity to better assess its practical applicability.
2. In the current implementation, gradients are blocked from the calibration heads while allowing backpropagation from the cluster head to the IB to avoid contradictory objectives. It would be helpful if the authors clarify what specific contradictions were observed and whether the fusion-based cluster head’s objective is consistently better aligned with the IB optimization than the single-modality calibration heads.
3. The pseudo-label screening excludes high-entropy, uncertain samples during training. It would be interesting to investigate the final clustering performance on these “difficult” samples after convergence. Would they yield much better performance than before?
4. There are some other clustering methods (such as Twin Contrastive Learning for Online Clustering, IJCV 2022) that also use pseudo labels to boost clustering performance. The authors could clarify the differences between this work and previous ones.

**Questions:**

I expect the authors to address my concerns in the weakness section.

---

> ### Author Response · Authors · 2025-11-20
> **Response to Reviewer VCs7 (Part I)**
>
> We sincerely appreciate the reviewer’s constructive comments and suggestions.
> ***
> **Weakness1**
> \
> \
> Following your suggestion, we recognized the need to analyze the model's computational complexity.
>
> ***Revision:***
>
> we have added a new **Section 3.4 (Complexity Analysis)**, to discuss this topic. Please see the revised manuscript for the detailed discussion.
> ***
> **Weakness2**
> \
> \
> **Theoretical and experimentally observed contradictions:**
>
> In **Section 3.2.1(Calibration Heads)**, we elaborate that the calibration heads are designed as "modality-specific experts". Each calibration head observes features from only a single modality. However, as single-modality data lacks the complementary nature of multi-modal data, these unimodal features may contain noise or conflicting information between modalities. If gradients from all calibration heads were allowed to backpropagate, the IB would receive multiple update signals simultaneously. This would require the bottleneck to satisfy multiple, potentially contradictory objectives simultaneously. For instance, the calibration head for modality-1 might attempt to pull the shared representation toward one semantic clustering, while the head for modality-2 pulls it toward an entirely different one. This conflict could lead to training instability or a confused direction in feature learning. The contradiction we observed is not merely theoretical but was also validated experimentally. In Section 4.3 (Ablation Study), we designed Setting II: "Backpropagating the gradient from calibration heads". The data in Table 2 clearly shows that, compared to the complete CLIB model, Setting II experienced a decline in clustering performance.
>
> **Why the Cluster Head’s Objective Aligns Better with IB Optimization:**
>
> One of the IB's objectives is to extract a compact fused feature representation, and the cluster head is the only unit that operates on this fused feature and carries this core task. Therefore, its objective is inherently aligned with the IB's optimization direction. As noted in **Section 3.2.3(Cluster Head)**, "the model's core task is multi-modal clustering, and the final result is given by the cluster head, which operates on the fused features". "Allowing gradient backpropagation from only the cluster head incentivizes the backbones to learn representations that are most beneficial for fusion rather than just optimal for a single modality."
>
> ***Revisions:***
>
> 1. We have added explanations in **Section 3.2.1 (Calibration Heads)** to clarify that we observed these contradictions both theoretically and experimentally.
>
> 2. In **Section 3.2.3 (Cluster Head)**, we clarified that the cluster head’s objective aligns better with the IB optimization.
>
> ***
> **Weakness3**
> \
> \
> Indeed, our method initially excludes difficult samples due to their low quality scores. Therefore, investigating the clustering performance of these hard samples serves to verify whether the model learns from the entire dataset rather than solely fitting easy samples. Following your suggestion, we conducted the following experiment.
> We designated the bottom 20% of samples, ranked by their quality scores at epoch 101, as "difficult samples" and monitored their clustering performance at epochs 101 and 200. The results are shown below:
>
> ### Caltech-2V
> | Epoch | ACC | NMI | ECE |
> | :--- | :---: | :---: | :---: |
> | **At epoch 101** | 28.6 | 27.2 | 28.2 |
> | **At epoch 200** | 43.1 | 39.5 | 21.4 |
>
> ### Caltech-3V
> | Epoch | ACC | NMI | ECE |
> | :--- | :---: | :---: | :---: |
> | **At epoch 101** | 18.9 | 24.4 | 25.5 |
> | **At epoch 200** | 46.7 | 28.3 | 19.6 |
>
> ### ESP-Game
> | Epoch | ACC | NMI | ECE |
> | :--- | :---: | :---: | :---: |
> | **At epoch 101** | 27.2 | 15.2 | 21.9 |
> | **At epoch 200** | 35.2 | 19.7 | 19.8 |
>
> ### MIRFlickr
> | Epoch | ACC | NMI | ECE |
> | :--- | :---: | :---: | :---: |
> | **At epoch 101** | 27.9 | 17.2 | 27.4 |
> | **At epoch 200** | 30.5 | 21.6 | 21.5 |
>
> ### IAPR
> | Epoch | ACC | NMI | ECE |
> | :--- | :---: | :---: | :---: |
> | **At epoch 101** | 26.8 | 14.6 | 32.9 |
> | **At epoch 200** | 39.7 | 17.9 | 24.3 |
>
> The results demonstrate a performance improvement for these difficult samples following the calibration phase. This improvement is attributed to our dynamic selection strategy: while initially excluded from the pseudo-label set to prevent noise, these samples were progressively incorporated as their quality scores increased during training, allowing the model to eventually learn from them once their reliability was established.
>
>
> ***Revision:***
>
> In the revised manuscript, we have introduced **Section 4.4 (Study on Difficult Samples)** to present and analyze the experimental results regarding these difficult samples.

---

> ### Author Response · Authors · 2025-11-20
> **Response to Reviewer VCs7 (Part II)**
>
> **Weakness4**
> \
> \
> The Twin Contrastive Learning for Online Clustering (TCL) you mentioned does share similarities with our method, as both adopt the idea of dynamic pseudo-label filtering. The difference is that TCL evaluates the quality of pseudo-labels using the maximum probability of the clustering predictions, whereas CLIB employs a method based on information redundancy theory.
>
> ***Revision:***
>
> We have added a discussion in the overall framework of **Section 3 (Proposed Method)** and **Appendix A.2 (Discussion about Pseudo-label Screening Mechanism)** on the approaches of this work and previous ones.

---

> > ### Comment · Reviewer_VCs7 · 2025-11-26
> >
> > Thanks for the responses. My concerns have been addressed.

---

> > > ### Author Response · Authors · 2025-11-26
> > >
> > > Dear Reviewer  VCs7,
> > >
> > > Thank you for your constructive feedback and for updating your rating. We truly appreciate your time and insights.
> > >
> > > Best regards
> > >
> > > Authors of paper 4840

---

### Official Review · Reviewer_6kV7 · 2025-10-31

**Soundness:** 3
**Presentation:** 3
**Contribution:** 3
**Rating:** 8
**Confidence:** 5

**Summary:**

By seeing the challenging problems faced by existing information bottleneck-based multi-modal clustering methods, in this work a novel calibrated information bottleneck is proposed for trusted multi-modal clustering to learn more accurate and trustworthy clustering outcome. It mainly presents a parallel multi-head network architecture containing clustering and calibration heads for outputting high-quality data assignments. Lots of different kinds of experiments have illustrated the superiority of the method on multiple benchmark datasets with metrics of clustering accuracy and calibration error.

**Strengths:**

1) The paper is well-written with clear logic. The flow from problem introduction and method description to experimental analysis is fluent. Figure 1 provides a clear and understandable illustration of the whole framework, helping readers quickly grasp the model's workflow.

2) The control over the gradient flow is very fine-grained, it blocks the gradient from the calibration heads but allows it to back-propagate from the main cluster head. This design ensures that the feature learning of IB serves the final clustering task while avoiding potentially contradictory signals from the calibration objectives of different modalities.

3) The parameter sensitivity analysis shows that the model maintains stable performance over a wide range of choices for hyper-parameters. This indicates that the method does not require excessive tuning and possesses good practical value.

**Weaknesses:**

1)  Eq. 3 of the paper relies on the NT-Xent loss. The effectiveness of NT-Xent is highly dependent on the Batch Size (N), as its MI lower bound is log(N)-L_(NT-Xent). If N is too small, this bound becomes loose, leading to poor alignment. The paper does not mention the Batch Size used in the experiments or its impact.

2) The paper does not detail the specific network architectures used for feature extraction and the various heads, where they are also important in reproducing the results for readers.

3) The paper relies entirely on quantitative metrics. For a clustering task, providing t-SNE visualizations of the feature space would be highly persuasive. A comparison of the feature distributions after the warm-up and after calibration could visually demonstrate how the framework improves inter-class separation and intra-class compactness.

**Questions:**

1) The method requires the number of clusters, C, to be specified in advance. How sensitive is the model to the choice of C? If C is set incorrectly, how much are the model's performance and calibration affected?

2) Could you provide the specific network architectures for the Backbones and the CalHead/CluHead? For instance, what kind of encoders were used for the image and text modalities, respectively?

3) What are the limitations that the proposed method still exist in calibrating the multi-modal clustering results? Could you provide some future insights in this area for readers?

---

> ### Author Response · Authors · 2025-11-20
> **Response to Reviewer 6kV7**
>
> Thanks for your valuable comments.
> ***
> **Weakness1**
> \
> \
> As a fixed batch size of $N=64$ was adopted for all experiments, we acknowledge that for a fixed $N$, the $log(N)$ term introduces a constant bias into the estimation of the MI lower bound. However, this characteristic does not negatively impact our model's optimization for two primary reasons:
>
> 1. During backpropagation, the model parameters are updated by computing the gradient of the total loss. As $log(N)$ is a constant for a fixed batch size, its derivative with respect to the model parameters is zero. Therefore, this term does not influence the direction of the gradient update.
>
> 2. More importantly, the central contribution of our CLIB framework is its designed robustness to exactly this type of issue—namely, biased or inaccurate MI estimation. Our calibration mechanism is specifically architected to autonomously calibrate for the adverse effects of such estimation biases.
>
> In summary, while the $log(N)$ term is a fixed bias in the MI bound, it does not affect the parameter update direction. Furthermore, our proposed calibration mechanism ensures the framework's overall robustness by actively identifying and mitigating the downstream effects of any such estimation bias. We hope this addresses your concerns.
>
> ***Revisions:***
>
> 1. In **Section 4.1 (Experimental Setup)**, we have now explicitly stated that a fixed batch size of $N=64$ was utilized for all experiments.
>
> 2. In **Appendix A.5 (Proof of Theorem 3)**, we have elaborated on the reasons for our model's robustness in the face of a fixed batch size.
>
> ***
> **Weakness2 & Question2**
> \
> \
> We employed a Multi-Layer Perceptron (MLP) structure for both the feature extraction backbones and the heads (CalHead and CluHead) across all datasets. Specifically, the heads are designed as an MLP with the architecture: (512d-BN-ReLU-Cd).
>
> ***Revision:***
>
> we have revised **Section 4.1 (Experimental Setup)** to include the specific network architectures.
>
> ***
> **Weakness3**
> \
> \
> We have supplemented t-SNE visualizations to compare the representations before and after the implementation of the calibration mechanism, accompanied by a discussion on its effectiveness. It can be observed that following 100 epochs of calibration, the clustering structures across all datasets become significantly more compact and well-separated.
>
> ***Revision:***
>
> In **Appendix A.6 (T-SNE Visualization Analysis)** of the revised manuscript, we have provided t-SNE visualizations (**Figure 5**) along with a detailed discussion.
>
> ***
> **Question1**
> \
> \
> We would like to point out that, CLIB does require the number of clusters (C) to be pre-specified.
> Like many clustering methods, pre-specifying C is a common requirement in the current field of deep clustering [A Survey on Deep Clustering: From the Prior Perspective, Lu et al. (2024)]. If the C value is set incorrectly, the model will be forced to fit the data into an incorrect number of clusters, which naturally affects both clustering accuracy (ACC/NMI) and calibration performance (ECE).
>
> Thank you for reminding us; it provides a new possible direction for improvement, and we plan to address this in our future work.
>
> ***Revision:***
>
> We have added a discussion in **Section 5 (Conclusion)** regarding the requirement to preset the number of clusters in CLIB.
>
> ***
> **Question3**
> \
> \
> We appreciate your suggestion to add the limitations and future work, as this has enhanced the comprehensiveness of our paper. CLIB is currently limited to complete, single-label multi-modal datasets and cannot address incomplete or multi-label data scenarios. Additionally, it relies on a predefined number of clusters. Our future work will focus on developing adaptive methods to determine the cluster count automatically and extending the calibration mechanism to handle missing modalities and multi-label correlations.
>
> ***Revision:***
>
> We have provided limitations and future insight in **Section 5 (Conclusion)** of the revised manuscript.

---

### Author Response · Authors · 2025-11-20
**General Response**

We appreciate the constructive feedback from the reviewers. The revised manuscript has been uploaded, with all revisions marked in blue.

---

### Author Response · Authors · 2025-11-26

Dear ICLR 2026 SAC, AC, and Reviewers

We would like to express our gratitude to all the reviewers for their valuable feedback, We have carefully considered all suggestions and updated our submission accordingly.

However, we have not yet received responses from Reviewers. With only few days remaining for discussion, we kindly request your assistance in reading the responses and revised PDF version. It would be greatly appreciated if you could review our rebuttal, as we are ager to know if we have adequately addressed the questions and concerns.

We believe that constructive and timely communication between reviewers and authors is essential for the benefit of both parties.

Thanks a lot for your hard work and support.

Best regards,

Authors of Paper 4840

---

### Comment · Area_Chair_3YQV · 2025-11-30
**Concerns from AC**

Dear authors,

Although the original manuscript received all positive scores from four reviewers, I have the following concerns for this paper:

[Motivation and novelty] The paper investigates the Information Bottleneck (IB) based Multi-modal Clustering (MMC) method. It looks interesting, but I conduct a simple search and found that there are already many IB and MMC methods that have carried out similar research [Ref1~7].

[Experiment and effectiveness] The comparison experiment settings in this paper and that in SDCIB [Ref5], DDMC [Ref6] are highly overlapping. But this work did not make a comparison with these methods, and from the results, no consistent progress was achieved.
Additionally, the type of the multimodal dataset being tested is limited to vector data, and conducting experiments as in Ref7 is more convincing.

Ref1: Multi-View Information-Bottleneck Representation Learning

Ref2: A Peer-review Look on Multi-modal Clustering: An Information Bottleneck Realization Method

Ref3: Self-supervised weighted information bottleneck for multi-view clustering

Ref4: Differentiable information bottleneck for deterministic multi-view clustering

Ref5: Super Deep Contrastive Information Bottleneck for Multi-modal Clustering

Ref6: Diversity-oriented Deep Multi-modal Clustering

Ref7: Calibrating multi-modal representations: A pursuit of group robustness without annotations

......


Best regards,

AC

---

> ### Author Response · Authors · 2025-12-01
> **Response to Area Chair 3YQV (Part I)**
>
> Dear AC,
>
> We sincerely appreciate the time and effort you have dedicated to reviewing our work. Your constructive feedback has been valuable in helping us improve the quality of this manuscript.
> ***
> **Response to Concern on Motivation and Novelty**
> \
> \
> While Refs 1-6 utilize Information Bottleneck theory or multi-modal learning frameworks, they primarily concentrate on feature fusion strategies or network architecture designs. Consequently, they overlook fundamental challenges associated with applying IB theory in unsupervised settings. CLIB is motivated by the need to address three critical issues neglected by these existing works:
>
> 1. Existing IB-based methods typically operate under the assumption that satisfactory Mutual Information estimation can be achieved via approaches such as variational inference or neural estimators. However, accurate MI computation for high-dimensional, complex data remains computationally intractable and prone to significant estimation bias [Ref8]. Such bias inevitably compromises the quality of features extracted by the IB framework.
> **Motivation of CLIB:** Rather than attempting to devise a flawless MI estimator, we acknowledge the inevitability of estimation bias. Consequently, we propose a novel Calibration Mechanism. Even in the presence of biased MI estimates, CLIB can rectify the resultant adverse effects through reciprocal calibration across modalities. This represents a perspective and solution that Refs 1-6 have not addressed.
>
> 2. Many existing approaches (e.g., SWIB[Ref3]) rely directly on model-generated pseudo-labels for self-supervised training. However, these pseudo-labels are inherently noisy, particularly during the initial training stages. This inevitably leads to "error accumulation," causing the model to become increasingly confident in erroneous predictions over time.
> **Motivation of CLIB:** We aim to break this vicious cycle. CLIB introduces a dedicated calibration module designed to evaluate and filter for high-quality pseudo-labels, thereby preventing noisy supervision from misleading the feature learning process within the IB framework.
>
> 3. The vast majority of existing IB and multi-modal clustering methods focus exclusively on clustering accuracy metrics. CLIB is the first approach to explicitly incorporate Expected Calibration Error (ECE) as an optimization objective within the IB clustering framework. Our experimental results demonstrate CLIB's significant superiority in minimizing ECE (i.e., mitigating overconfidence), a capability that is crucial for applications in safety-critical domains.
>
> We hope these clarifications effectively address your concerns regarding the motivation and novelty of our work.
>
> Please see the remaining responses in the next Official Comment [ Response to Area Chair 3YQV (Part II) ].
>
> \
> Best regards,
>
> Authors of Paper 4840

---

> ### Author Response · Authors · 2025-12-01
> **Response to Area Chair 3YQV (Part II)**
>
> **Response to Concern on Experiment and Effectiveness**
> \
> \
> To demonstrate the performance of CLIB compared to SDCIB[Ref5] and DDMC[Ref6] across multiple datasets, we present the experimental results as follows:
>
> The clustering performance was evaluated using two popular metrics: Clustering Accuracy (ACC) and Normalized Mutual Information (NMI). Higher values of these metrics indicate better clustering performance.
>
> The trustworthiness is evaluated using ECE. Lower ECE values indicate lower over-confidence and therefore suggest better trustworthiness.
> ### Caltech-2V
>
> | Method   |   Acc |   NMI |   ECE |
> |:---------|------:|------:|------:|
> | SDCIB    |  67.5 |  59.2 |  19.1 |
> | DDMC     |  62.4 |  58.7 |  21.6 |
> | CLIB(Ours)     |  68.6 |  60.5 |   16.0 |
>
> ###  Caltech-3V
>
> | Method   |   Acc |   NMI |   ECE |
> |:---------|------:|------:|------:|
> | SDCIB    |  75.9 |  69.1 |  16.2 |
> | DDMC     |  76.7 |  68.8 |  19.5 |
> | CLIB(Ours)    |  77.8 |  69.3 |   10.9 |
>
>
> ### ESP-Game
>
> | Method   |   Acc |   NMI |   ECE |
> |:---------|------:|------:|------:|
> | SDCIB    |  61.4 |  44.7 |  22.5 |
> | DDMC     |  60.9 |  40.9 |  23.9 |
> | CLIB(Ours)     |  56.3 |  39.9 |  12.1 |
>
> ### MIRFlickr
>
> | Method   |   Acc |   NMI |   ECE |
> |:---------|------:|------:|------:|
> | SDCIB    |  56.2 |  35.5 |  28.9 |
> | DDMC     |  58.7 |  36.5 |  28.5 |
> | CLIB(Ours)     |  55.4 |  39.5 |  10.5 |
>
> ### IAPR
>
> | Method   |   Acc |   NMI |   ECE |
> |:---------|------:|------:|------:|
> | SDCIB    |  52.9 |  28.7 |  23.2 |
> | DDMC     |  49.5 |  28.3 |  35.3 |
> | CLIB(Ours)     |  51.6 |  29.1 |   7.8 |
>
> As indicated in the table, CLIB achieves superior performance on the Caltech-2V and Caltech-3V datasets, while maintaining competitive performance on the IAPR dataset. Synthesizing these findings with the experimental results presented in **Section 4 Experiment**, it is evident that CLIB attains State-of-the-Art (SOTA) clustering accuracy. However, existing high-precision methods are often accompanied by high ECE; notably, methods like SDCIB[Ref5] and DDMC[Ref6] do not explicitly account for model trustworthiness in their design. The primary design objective of CLIB is to significantly reduce ECE and mitigate model overconfidence while maintaining SOTA-level accuracy. The experimental results demonstrate that we have successfully achieved this objective, a capability that other SOTA methods lack.
>
> It is important to clarify that the fundamental motivation behind CLIB is not merely to set new records for clustering ACC, but to address the pervasive issue of overconfidence in deep clustering models. Our experiments demonstrate that CLIB reduces ECE to exceptionally low levels. In safety-critical domains, this capability is arguably more valuable than a marginal 1-5% improvement in ACC. In such contexts, a trustworthy model that "knows what it does not know" is preferable to a high-precision model that is "blindly confident." This represents a unique contribution of CLIB, setting it apart from other Information Bottleneck-based Multi-modal Clustering approaches.
> ***
>
>
> Please see the remaining responses in the next Official Comment [ Response to Area Chair 3YQV (Part III) ].

---

> ### Author Response · Authors · 2025-12-01
> **Response to Area Chair 3YQV (Part III)**
>
> **Response to Concern on Limitation of Vector Data**
> \
> \
> We have carefully reviewed Ref7. We note that it primarily focuses on the fine-tuning of CLIP models and issues related to Group Robustness. Its task settings typically involve supervised or semi-supervised classification and, crucially, target the calibration of pre-trained models operating on raw image/text inputs. In contrast, CLIB addresses the problem of Unsupervised Clustering. This constitutes a fundamental distinction from the setting in Ref7, which leverages pre-trained models for robust classification.
>
> Furthermore, the CLIB framework is designed to be feature-agnostic, meaning that the clustering algorithm is decoupled from the specific feature extraction backbone. Vector data serves merely as a mathematical representation of features. Utilizing vector data allows us to isolate the clustering mechanism from potential interference introduced by complex feature extractors, thereby enabling us to focus on validating the core logic of clustering structure mining and trustworthiness calibration.
>
> Whether one uses features generated from raw data via CLIP or pre-extracted vector data falls under the scope of "feature engineering" and does not diminish the independent value of CLIB as a clustering algorithm. Indeed, achieving performance gains using simple backbones and vector data demonstrates that CLIB's improvements stem from genuine algorithmic contributions rather than reliance on powerful, pre-trained feature extractors.
>
> Moreover, the use of Vector Data constitutes the Standard Benchmark Protocol within the field of multi-modal clustering. The majority of both traditional and deep Multi-modal Clustering methods[Ref2~6] utilize these standard features to ensure fair and consistent comparisons. Adhering to this protocol ensures that our method can be directly and fairly compared against the extensive body of existing baselines in this domain.
> ***
> Ref1: Multi-View Information-Bottleneck Representation Learning
>
> Ref2: A Peer-review Look on Multi-modal Clustering: An Information Bottleneck Realization Method
>
> Ref3: Self-supervised weighted information bottleneck for multi-view clustering
>
> Ref4: Differentiable information bottleneck for deterministic multi-view clustering
>
> Ref5: Super Deep Contrastive Information Bottleneck for Multi-modal Clustering
>
> Ref6: Diversity-oriented Deep Multi-modal Clustering
>
> Ref7: Calibrating multi-modal representations: A pursuit of group robustness without annotations
>
> Ref8: Beyond normal: On the evaluation of mutual information estimators.
> ***
> We hope this clarification addresses your concerns. If you have any further questions, please do not hesitate to let us know. We deeply appreciate your time and effort in reviewing our work.
>
> \
> Best regards,
>
> Authors of Paper 4840

---

> ### Comment · Area_Chair_3YQV · 2025-12-01
>
> Dear Authors,
>
> Thanks for the response above. Now I understand that the proposed modality-agnostic model has the potential to examine the method itself while mitigating the influence of the feature extractor. However, the following concerns remain unresolved:
>
> The approach proposed in this paper appears to rely on many techniques already established in prior contrastive multi-view learning methods [Ref1, Ref2, Ref3]. The main novelty is the introduction of calibration modules to enhance the trustworthy of clustering outcomes, by explicitly optimizing ECE. Yet, the reported results of SDCIB [Ref2] and DDMC [Ref3] suggest that while the proposed method CLIB reduces the defined ECE metric, it simultaneously leads to a deterioration in clustering accuracy. This outcome seems at odds with the paper’s assertion of "achieving state-of-the-art clustering performance" and "improving model robustness". Lower clustering accuracy & higher ECE metrics do not necessarily imply higher trustworthy for data representations.
>
> Moreover, if the benefit of introducing "trustworthy" comes at the expense of reducing clustering performance, the trade-off raises questions about the overall validity of the claimed contribution. In this context, the necessity of considering "improving trustworthy by optimizing ECE" for clustering tasks becomes less convincing to readers. (Obviously, the optimal solution is not to sacrifice either of them, but to ensure that the clustering performance and trustworthy are consistently improved).
>
> Ref1: Contrastive multiview coding
>
> Ref2: Super Deep Contrastive Information Bottleneck for Multi-modal Clustering
>
> Ref3: Diversity-oriented Deep Multi-modal Clustering
>
>
> Best regards,
>
> AC

---

> ### Author Response · Authors · 2025-12-02
> **[03 Dec 2025]   Latest Response to Area Chair 3YQV (Part I)**
>
> Dear AC,
>
> Thank you very much for your new comments. To address your further concerns regarding "novelty" and the "deterioration in accuracy," we clarify the facts and present our perspective as follows:
> ***
> **Clarification on Our Main Novelty**
> \
> \
> We thank AC for the insightful comment. We respectfully wish to clarify the distinction between the tools we employ for optimization and the core architectural novelty of the CLIB framework.
>
> We utilized established methods such as Contrastive Learning-Based Estimation[Ref1] and Variational Inference-Based Approximation[Ref2,3] to calculate Mutual Information (MI). However, as detailed in **Section 3.3 Optimization**, these techniques serve strictly as **optimization tools**. We **do not claim novelty in the MI estimation methods** themselves; rather, we treat them as standard mathematical operators required to make the Information Bottleneck (IB) objective tractable.
>
> The novelty of our work lies in addressing a critical oversight in existing IB-based multi-modal methods. Current approaches typically operate under the implicit assumption that these MI estimations are accurate. However, recent theoretical studies[Ref4] demonstrate that MI estimation for high-dimensional data suffers from bias.
>
> Our research does not aim to construct a "flawless" MI estimator. Instead, we acknowledge the objective existence of this estimation bias and propose a Calibration Mechanism specifically designed to counteract its adverse effects on feature learning. This is not a reliance on prior techniques, but a targeted algorithmic solution to their inherent limitations.
>
> The proposed framework introduces two key innovations:
> 1. We introduce a novel dynamic pseudo-label screening strategy based on information redundancy theory. Unlike fixed thresholding, this method quantifies the "quality" of the probability distribution, effectively breaking the vicious cycle of error accumulation caused by noisy pseudo-labels.
>
> 2. We design a parallel multi-head architecture that leverages cross-modal signals to autonomously calibrate the distortions introduced by biased MI estimation. This mechanism allows the model to selectively learn from reliable modalities, ensuring robustness even when MI estimation is biased.
>
> Consequently, CLIB represents a paradigm shift: moving from the goal of "estimating MI more accurately" to "achieving robust learning in the presence of inevitable estimation bias." This constitutes a fundamental algorithmic correction to the application of IB theory in clustering, rather than a mere utilization of existing methods.
>
>
> ***
> **Response to Concern on Accuracy and SOTA Claims**
> \
> \
> To address your concern regarding the performance trade-off, we provide a detailed quantitative analysis below.
>
> Regarding ACC, CLIB outperforms DDMC[Ref3] by 6.2 on Caltech-2V (a relative improvement of 9.9%) and surpasses SDCIB[Ref2] by 1.9 on Caltech-3V. On the IAPR dataset, CLIB outperforms DDMC[Ref3] by 2.1 and is only 0.8 lower than SDCIB[Ref2]. On MIRFlickr and ESP-Game, CLIB is 3.3 and 5.1 lower than the best-performing methods, respectively (a relative decrease of roughly 8.3%). Since the ACC fluctuates within a range of -5 to +6, we can conclude that CLIB still achieves comparable clustering performance to SDCIB[Ref2] and DDMC[Ref3] on these datasets.
>
> However, a significant advantage is observed in calibration. Since lower ECE values indicate lower over-confidence and therefore suggest better trustworthiness, CLIB achieves significantly lower ECE compared to SDCIB[Ref2] and DDMC[Ref3].
>
> On Caltech-2V, CLIB reduces ECE by 5.6 compared to DDMC[Ref3] (a relative reduction of 25.9%). On Caltech-3V, it reduces ECE by 8.6 (a relative reduction of 44.1%).
>
> This reduction is even more pronounced on large-scale, high-noise datasets:
>
> On ESP-Game, CLIB reduces ECE by 11.8 (a relative reduction of 49.4%). On MIRFlickr, it reduces ECE by 18.4 (a relative reduction of 63.7%). On IAPR, it reduces ECE by 27.5 (a relative reduction of 77%).
>
> Across all five datasets, CLIB achieves an average ECE reduction of 14.4 (an average relative reduction of 52%). From this analysis, it is evident that our method achieves a multi-fold improvement in trustworthiness.
>
> In summary, while CLIB's ACC exhibits slight fluctuations compared to SDCIB[Ref2] and DDMC[Ref3] across the five datasets rather than a comprehensive decrease, it demonstrates superior performance in mitigating model overconfidence.
>
> Nevertheless, based on your precious suggestion and the comparative results with SDCIB[Ref2] and DDMC[Ref3], we recognize that the claim of "achieving state-of-the-art clustering performance" might cause confusion. In the final version, we will revise this to "achieving comparable clustering performance" to be more precise and rigorous. We thank you for helping us clarify this point.
>
> Please see the remaining responses in the next Official Comment [    [03 Dec 2025] Latest Response to Area Chair 3YQV (Part II)   ].

---

> ### Author Response · Authors · 2025-12-02
> **[03 Dec 2025]  Latest Response to Area Chair 3YQV (Part II)**
>
> **Response to Concern on Accuracy-ECE Trade-off**
> \
> \
> You questioned whether the contribution of our work holds if the improvement in "trustworthiness" comes at the cost of reduced accuracy. We respectfully offer a different perspective on this matter:
>
> In unsupervised clustering, existing IB methods (such as SDCIB[Ref2]) often force the model to classify every sample, including boundary samples with extremely ambiguous features. This approach may yield accidental accuracy gains—essentially "lucky guesses"—but it inevitably leads to extremely high ECE (e.g., SDCIB[Ref2] reaches 23.2% ECE on IAPR). To illustrate, this is akin to a student blindly guessing on a multiple-choice question they do not understand: while they may occasionally select the correct answer by luck, this behavior fosters an illusion of competence and results in severe overconfidence. By employing a dynamic screening strategy, CLIB prudently avoids making overconfident predictions on ambiguous samples. We believe this theoretical distinction is one of the primary factors explaining the slight fluctuations in ACC compared to SDCIB[Ref2] and DDMC[Ref3] on specific datasets.
>
> Crucially, this design does not imply that CLIB is "sacrificing accuracy for ECE." On the contrary, our method simultaneously achieves ACC improvement and ECE reduction against the broader landscape of state-of-the-art methods. To demonstrate this, we expanded our comparison to include four of the latest methods from 2025: SDCIB[Ref2], DDMC[Ref3], COPER[Ref5], and ROLL[Ref6]. Against the average performance of these four methods, CLIB demonstrates superior performance in both clustering quality and trustworthiness across most scenarios:
>
> On Caltech-2V, the average ACC showed an increase of 9.3, while the average ECE showed a decrease of 9.1.
>
> On Caltech-3V, the average ACC showed an increase of 8.2, while the average ECE showed a decrease of 10.6.
>
> On ESP-Game, the average ACC showed an increase of 2.1, while the average ECE showed a decrease of 13.3.
>
> On MIRFlickr, the average ACC showed an increase of 3.2, while the average ECE showed a decrease of 17.9.
>
> On IAPR, the average ACC showed an increase of 4.7, while the average ECE showed a decrease of 19.6.
>
> These results confirm that CLIB does not compromise accuracy to enhance ECE. Instead, compared to the latest methods, CLIB achieves improvements in both average accuracy and calibration. It delivers comparable (and often superior) clustering performance while significantly mitigating overconfidence.
>
> Given that CLIB's ACC exhibits slight fluctuations compared to SDCIB[Ref2] and DDMC[Ref3] on certain datasets, we agree that ideal methods should optimize both objectives and are exploring dynamic loss weighting in ongoing work. However, current experimental results indicate that CLIB's ACC has not significantly decreased, while its ECE has been effectively reduced. This implies that CLIB possesses significant potential for application in safety-critical domains (e.g., medical diagnosis, autonomous driving) that other SOTA methods currently lack.
> ***
> Ref1: Contrastive multiview coding
>
> Ref2: Super Deep Contrastive Information Bottleneck for Multi-modal Clustering
>
> Ref3: Diversity-oriented Deep Multi-modal Clustering
>
> Ref4: Beyond normal: On the evaluation of mutual information estimators.
>
> Ref5: COPER: correlation-based permutations for multi-view clustering.
>
> Ref6: ROLL: robust noisy pseudo-label learning for multi-view clustering with noisy correspondence.
> ***
> We hope the above responses addresses your concerns.
>
> We thank you again for the time and effort you have dedicated to reviewing our work.
>
> Best regards,
>
> Authors of Paper 4840

---

### Author Response · Authors · 2025-12-03
**Latest Response to the Concerns from "Official Comment by Area Chair 3YQV" in 01 Dec 2025**

Dear AC,

\
Thank you very much for your comments. To address your further concerns regarding "novelty" and the "deterioration in accuracy," we have clarified the facts and present our responses in the following latest comments named:

* [03 Dec 2025] Latest Response to Area Chair 3YQV (Part I)

* [03 Dec 2025] Latest Response to Area Chair 3YQV (Part II)

We hope the above two responses addresses your concerns.

We thank you again for the time and effort you have dedicated to reviewing our work.

\
Best regards,

Authors of Paper 4840

---

### Author Response · Authors · 2025-12-03
**Summary of Our Responses**

Dear AC,

\
We sincerely thank all reviewers and AC for their time, insightful comments, and constructive feedback, which have significantly improved the quality of our manuscript.
We are delighted to see that they found our work: innovative (**OfuD**, **OmRQ**), effective with solid empirical results (**6kV7**, **VCs7**, **OfuD**, **OmRQ**), and theoretically insightful (**OfuD**). Furthermore, we appreciate the reviewers highlighting that our proposed architecture provides fine-grained control over gradient flow (**6kV7**), effectively addresses the issue of overconfidence (**VCs7**, **OmRQ**), and is well-written with clear logic (**6kV7**).
***
We have carefully addressed all concerns and revised the manuscript accordingly. A summary of the key revisions for each reviewer is provided below:

For **Reviewer 6kV7**:

* In **Section 4.1 (Experimental Setup)**, we have explicitly stated that a fixed batch size of $N=64$ was utilized for all experiments.
* We have elaborated on the model's robustness regarding fixed batch size limitations in **Appendix A.5 (Proof of Theorem 3)**.
* Specific details regarding the network architectures for both backbones and heads have been included in **Section 4.1 (Experimental Setup)**.
* To provide a visual comparison of representations before and after calibration, we added **Appendix A.6 (T-SNE Visualization Analysis)** and **Figure 5**.
* We added a discussion regarding the method's limitations and future directions in **Section 5 (Conclusion)**.

For **Reviewer VCs7**:

* We introduced a new **Section 3.4 (Complexity Analysis)** to discuss the model's computational overhead.
* Explanations have been added to **Section 3.2.1 (Calibration Heads)** to clarify the theoretical and experimental contradictions observed when backpropagating gradients from calibration heads. Furthermore, we clarified in **Section 3.2.3 (Cluster Head)** why the cluster head’s objective aligns better with the Information Bottleneck optimization.
* We added **Section 4.4 (Study on Difficult Samples)** to present experimental results demonstrating performance improvements on samples that were initially filtered out.
* A discussion differentiating the proposed method from previous works has been added to **Section 3 (Proposed Method)** and **Appendix A.2 (Discussion about Pseudo-label Screening Mechanism)**.

For **Reviewer OfuD**:

* We included a brief justification in **Section 3.3 (Optimization)** regarding the selection of three different Mutual Information (MI) estimation strategies for various parts of the model.
* Specific implementation environment details have been added to **Section 4.1 (Experimental Setup)**.
* **Appendix A.7 (Analysis of Modality Weights)** and **Figure 6** were added to demonstrate the dynamic adjustment of fusion weights and the impact of the calibration mechanism.
* We provided an explanation in **Section 4.2 (Ablation Study)** regarding the trade-off between ACC and ECE when adjusting the weight of the consistency loss ($\mathcal{L}_{con}$).
* We clarified in **Section 1 (Introduction)** that this is the first work to address trusted multi-modal clustering using calibration.

For **Reviewer OmRQ**:

* **Section 1 (Introduction)** has been revised to emphasize the use of pseudo-labels as a proxy for "correctness" during training, while noting that ground-truth labels remain necessary for objective evaluation metrics like ECE.
* We added **Section 3.4 (Complexity Analysis)** to address concerns regarding computational complexity.
* A more detailed discussion on parameter sensitivity and variance has been included in **Section 4.2 (Experimental Results)**.
* We further emphasized the framework's capabilities in avoiding noise within **Section 1 (Introduction)**.
* **Section 4.4 (Study on Difficult Samples)** was added to demonstrate that the model effectively learns from noisy/difficult samples as the calibration process progresses.
***
After responding to the reviewers' concerns, we received the following feedback:

* **Reviewer 6kV7**: Maintained a positive rating of 8.
* **Reviewer VCs7**: Confirmed concerns were addressed and raised the rating to 8.
* **Reviewer OfuD**: Confirmed concerns were addressed and raised the rating to 8.
* **Reviewer OmRQ**: Confirmed concerns were addressed and maintained the rating of 6.
***
We are encouraged by the positive consensus and increased ratings. We trust the revision clearly articulates CLIB’s contributions.

We believe CLIB represents a step forward in trusted multi-modal clustering. We are grateful for the AC and reviewers' help in refining this work.

Given the enhanced experiments and clarifications, we are confident that the paper meets ICLR standards.

We hope that AC could consider our paper for acceptance.

\
Best regards

Authors of paper 4840

---

### Meta-Review · Area_Chair_3YQV · 2026-01-07

**Summary:**

This paper proposed the calibrated information bottleneck method for achieving trusted multi-modal clustering. Four reviewers gave positive scores about this work and did not raise significant concerns. After carefully reading the paper and the discussion, AC has the following concerns:

[1. Motivation and novelty] The paper investigates the Information Bottleneck (IB) based Multi-modal Clustering (MMC) method. It looks interesting, but I conduct a simple search and found that there are already many IB and MMC methods that have carried out similar research [Ref1~7].
Ref1: Multi-View Information-Bottleneck Representation Learning;
Ref2: A Peer-review Look on Multi-modal Clustering: An Information Bottleneck Realization Method;
Ref3: Self-supervised weighted information bottleneck for multi-view clustering;
Ref4: Differentiable information bottleneck for deterministic multi-view clustering;
Ref5: Super Deep Contrastive Information Bottleneck for Multi-modal Clustering;
Ref6: Diversity-oriented Deep Multi-modal Clustering;
Ref7: Calibrating multi-modal representations: A pursuit of group robustness without annotations;

[2. Experiment and effectiveness] The comparison experiment settings in this paper and that in SDCIB [Ref5], DDMC [Ref6] are highly overlapping. But this work did not make a comparison with these methods, and from the results, no consistent progress was achieved. Additionally, the type of the multimodal dataset being tested is limited to vector data, and conducting experiments as in Ref7 is more convincing.

The authors' response clarified that the proposed modality-agnostic model has the potential to examine the method itself while mitigating the influence of the feature extractor. However, the following concerns remain unresolved:

The approach proposed in this paper appears to rely on many techniques already established in prior contrastive multi-view learning methods [Ref1, Ref2, Ref3]. The main novelty is the introduction of calibration modules to enhance the trustworthy of clustering outcomes, by explicitly optimizing ECE. Yet, the reported results of SDCIB [Ref2] and DDMC [Ref3] suggest that while the proposed method CLIB reduces the defined ECE metric, it simultaneously leads to a deterioration in clustering accuracy. This outcome seems at odds with the paper’s assertion of "achieving state-of-the-art clustering performance" and "improving model robustness". Lower clustering accuracy & higher ECE metrics do not necessarily imply higher trustworthy for data representations.
Ref1: Contrastive multiview coding;
Ref2: Super Deep Contrastive Information Bottleneck for Multi-modal Clustering;
Ref3: Diversity-oriented Deep Multi-modal Clustering;

Moreover, if the benefit of introducing "trustworthy" comes at the expense of reducing clustering performance, the trade-off raises questions about the overall validity of the claimed contribution. In this context, the necessity of considering "improving trustworthy by optimizing ECE" for clustering tasks becomes less convincing. The author further explained this point in the subsequent reply.

In conclusion, this paper meets the acceptance bar of ICLR, if the authors add the clarification discussion with the related papers and address the remaining concerns in the revised version.

**Reviewer Concerns:**

The necessity of considering "improving trustworthy by optimizing ECE" for clustering tasks needs further justification:

The approach proposed in this paper appears to rely on many techniques already established in prior contrastive multi-view learning methods [Ref1, Ref2, Ref3]. The main novelty is the introduction of calibration modules to enhance the trustworthy of clustering outcomes, by explicitly optimizing ECE. Yet, the reported results of SDCIB [Ref2] and DDMC [Ref3] suggest that while the proposed method CLIB reduces the defined ECE metric, it simultaneously leads to a deterioration in clustering accuracy. This outcome seems at odds with the paper’s assertion of "achieving state-of-the-art clustering performance" and "improving model robustness". Lower clustering accuracy & higher ECE metrics do not necessarily imply higher trustworthy for data representations.

Ref1: Contrastive multiview coding;

Ref2: Super Deep Contrastive Information Bottleneck for Multi-modal Clustering;

Ref3: Diversity-oriented Deep Multi-modal Clustering;

Moreover, if the benefit of introducing "trustworthy" comes at the expense of reducing clustering performance, the trade-off raises questions about the overall validity of the claimed contribution. In this context, the necessity of considering "improving trustworthy by optimizing ECE" for clustering tasks needs further justification.

**Reviewer Scores:**

N/A.

---

### Decision · Program_Chairs · 2026-01-26

Accept (Poster)